# *Candidatus* Alkanophaga archaea from Guaymas Basin hydrothermal vent sediment oxidize petroleum alkanes

Hanna Zehnle [1,2,3] ✉, Rafael Laso-Pérez [2,4,7], Julius Lipp [2], Dietmar Riedel[5], David Benito Merino [1,3], Andreas Teske [6] & Gunter Wegener [1,2] ✉

Methanogenic and methanotrophic archaea produce and consume the greenhouse gas methane, respectively, using the reversible enzyme methyl-coenzyme M reductase (Mcr). Recently, Mcr variants that can activate multicarbon alkanes have been recovered from archaeal enrichment cultures. These enzymes, called alkyl-coenzyme M reductase (Acrs), are widespread in the environment but remain poorly understood. Here we produced anoxic cultures degrading mid-chain petroleum $n$-alkanes between pentane ($C_5$) and tetradecane ($C_{14}$) at 70 °C using oil-rich Guaymas Basin sediments. In these cultures, archaea of the genus *Candidatus* Alkanophaga activate the alkanes with Acrs and completely oxidize the alkyl groups to $CO_2$. *Ca*. Alkanophaga form a deep-branching sister clade to the methanotrophs ANME-1 and are closely related to the short-chain alkane oxidizers *Ca*. Syntrophoarchaeum. Incapable of sulfate reduction, *Ca*. Alkanophaga shuttle electrons released from alkane oxidation to the sulfate-reducing *Ca*. Thermodesulfobacterium syntrophicum. These syntrophic consortia are potential key players in petroleum degradation in heated oil reservoirs.

In deep seafloor sediments, pressure and heat transform buried organic matter into complex hydrocarbon mixtures, forming natural gas and crude oil[1,2]. *n*-Alkanes (hereafter referred to as 'alkanes') constitute a major fraction of these mixtures[3] and become energy-rich substrates for microorganisms[4] in habitable anoxic zones. Sulfate-reducing bacteria (SRB) oxidize alkanes ≥ propane ($C_3$ alkane)[5,6] after activation via fumarate addition through alkylsuccinate synthases[7]. Archaea possess a different mechanism for anaerobic alkane degradation based on reversal of the methanogenesis pathway. This mechanism was first revealed in anaerobic methanotrophic archaea (ANME)[8,9], which activate methane to methyl-coenzyme M (methyl-CoM) via the key enzyme of methanogenesis methyl-coenzyme M reductase (Mcr)[10]. Recently cultured archaea oxidize non-methane alkanes analogously to ANME, as a first step activating the alkanes to alkyl-CoMs via divergent variants of the Mcr, termed alkyl-CoM reductases (Acrs)[11]. *Candidatus* Argo-archaeum ethanivorans[12], *Ca*. Ethanoperedens thermophilum[13] and *Ca*. Syntrophoarchaeum spp.[14] oxidize short-chain gaseous alkanes ($C_2$-$C_4$), while *Ca*. Methanoliparum spp., enriched from oil-rich environments, oxidize long-chain alkanes ($\geq C_{16}$)[15]. Similar to most ANME, the short-chain alkane-oxidizing archaea lack respiratory pathways and shuttle the electrons from alkane oxidation to partner SRB[13,14,16,17]. In contrast, *Ca*. Methanoliparum encodes a canonical Mcr in addition to the Acr, with which it couples alkane oxidation to methanogenesis in a single cell[15].

[1]Max Planck Institute for Marine Microbiology, Bremen, Germany. [2]MARUM, Center for Marine Environmental Sciences, University of Bremen, Bremen, Germany. [3]Faculty of Geosciences, University of Bremen, Bremen, Germany. [4]Systems Biology Department, Centro Nacional de Biotecnología (CNB-CSIC), Madrid, Spain. [5]Max Planck Institute for Multidisciplinary Sciences, Göttingen, Germany. [6]Department of Earth, Marine and Environmental Sciences, University of North Carolina at Chapel Hill, Chapel Hill, NC, USA. [7]Present address: Biogeochemistry and Microbial Ecology Department, Museo Nacional de Ciencias Naturales (MNCN-CSIC), Madrid, Spain. ✉e-mail: hzehnle@mpi-bremen.de; gwegener@marum.de

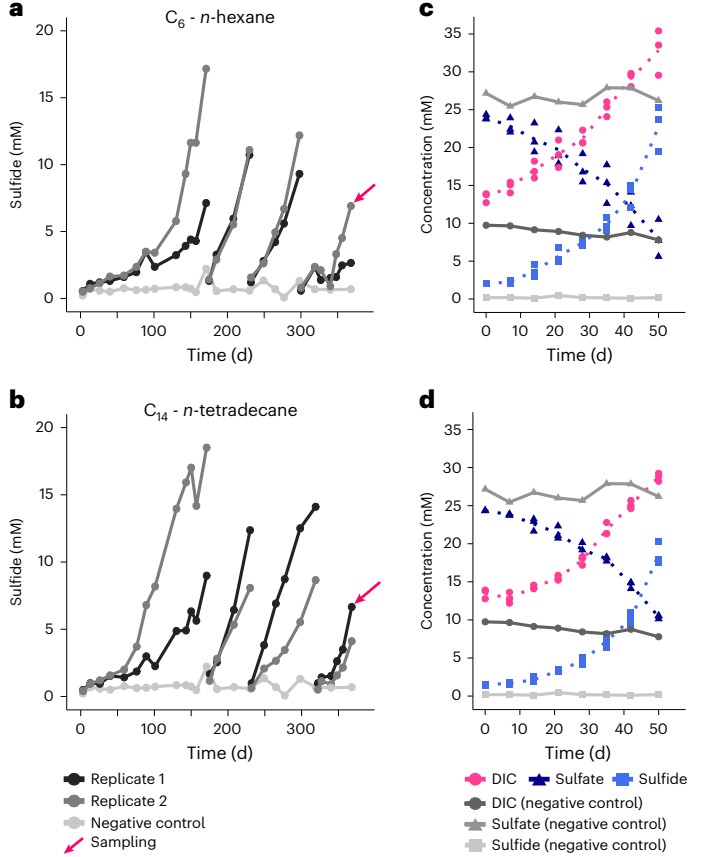

**Fig. 1 | Metabolic activity in anaerobic petroleum alkane-oxidizing cultures at 70 °C. a,b**, Formation of sulfide over time in *n*-hexane (C$_6$) (**a**) and *n*-tetradecane (C$_{14}$) (**b**) cultures. Gaps in concentration profiles indicate dilution events. Arrows mark sampling for metagenomic and transcriptomic analyses. **c,d**, Concentrations of dissolved inorganic carbon (DIC), sulfate and sulfide in the C$_6$ (**c**) and C$_{14}$ (**d**) cultures, and in abiotic controls. For the cultures, three replicate samples were measured, with arithmetic mean shown as a dotted line.

Anaerobic archaea capable of petroleum alkane (C$_5$-C$_{15}$) oxidation via Acrs were unknown. These alkanes are the major constituents of gasoline and kerosene[18,19], and of high ecological relevance because of their toxicity[20,21]. Lately, many *acr* genes with unknown function have been recovered from environmental metagenomes, especially from hot springs[22–24]. We hypothesized that yet uncultured thermophilic archaea could activate petroleum alkanes via Acrs. We aimed to enrich such archaea from heated oil-rich sediment from the hydrothermal vent site Guaymas Basin (Gulf of California, Mexico)[25]. We obtained eight enrichment cultures thriving at 70 °C, in which alkanes from C$_5$-C$_{14}$ were oxidized in combination with sulfate reduction. Analyses of these cultures via omics approaches and physiological tests revealed that a sister clade of ANME-1, *Ca.* Alkanophaga, was oxidizing the alkanes after activation via Acrs coupled to sulfate reduction by a partner *Thermodesulfobacterium*. Such consortia potentially contribute to souring in deeply buried, heated oil reservoirs.

## Results

### Thermophilic microorganisms thrive on petroleum alkanes

Anoxic slurries produced from heated sediment collected at the hydrothermal vent complex Cathedral Hill in the Southern Trough of the Guaymas Basin (Extended Data Fig. 1a–d) were amended with petroleum alkanes (C$_5$-C$_{14}$) as sole carbon and electron source and sulfate as electron acceptor, and incubated at 70 °C. Within 3–7 months, the

slurries produced >10 mM sulfide. Sulfide production was accompanied by dissolved inorganic carbon (DIC) production and sustained during dilution steps (Fig. 1 and Extended Data Fig. 2), yielding effectively sediment-free cultures after the third dilution. Cultures, except the considerably slower C$_5$ culture, doubled within 13–40 d (Supplementary Table 1).

According to the general formula

$$C_nH_{2n+2} + (0.75n + 0.25)SO_4^{2-} \rightarrow nHCO_3^- +$$
$$(0.75n + 0.25)HS^- + H_2O + (0.25n - 0.25)H^+, \tag{1}$$

the ratio of DIC production to sulfate reduction is ~1.25–1.30 in case of complete alkane oxidation. In two representative cultures (C$_6$ and C$_{14}$), this ratio was slightly lower, with 1.21 ± 0.22 in the C$_6$ culture and 1.09 ± 0.04 in the C$_{14}$ culture. These values suggest that around 10% (C$_6$) and 35% (C$_{14}$) of the carbon released from alkane oxidation is assimilated into biomass (Supplementary Table 2).

### *Ca.* Alkanophagales archaea are abundant in the cultures

We reconstructed two high-quality archaeal metagenome-assembled genomes (MAGs) from the cultures (Supplementary Table 3): MAG 4, abundant in the C$_5$-C$_7$ cultures and MAG 1, abundant in the C$_8$-C$_{14}$ cultures (Fig. 2a and Supplementary Table 4). Both MAGs were rare (relative abundances ≤0.1%) in the original slurry (Extended Data Fig. 1e,f). The in situ temperatures of the studied sediment (Extended Data Fig. 1d), which captured only the upper sediment layer up to 30 cm depth, probably did not reach the optimal growth temperatures of the two organisms. Both MAGs recruited up to 39% (MAG 4) and 5% (MAG 1) of raw reads in deeper, hotter layers of the Guaymas Basin[26] (Supplementary Table 5).

MAGs 1 and 4 represent two species within one genus (average nucleotide identity (ANI) 81.5%) and belong to the same genus as the previously published MAG ANME-1 B39_G2 reconstructed from Guaymas Basin sediments (ANIs: MAG 1-ANME-1 B39_G2 98.8% and MAG 4-ANME-1 B39_G2 80.8%)[27]. The name *Ca.* Alkanophagales was recently proposed for the clade represented by ANME-1 B39_G2 on the basis of its genomic content which hinted at a capacity for multicarbon alkane metabolism[11,27]. MAGs 1 and 4 form a clade diverging at the root of ANME-1 and next to *Ca.* Syntrophoarchaeum, together forming the class Syntrophoarchaeia (Fig. 2b).

Visualization of the organisms revealed mixed aggregates of archaea of the *Ca.* Alkanophagales clade and bacteria (Fig. 2c–f). These associations resemble those of short-chain alkane-oxidizing cultures[13,14,16], suggesting that archaea oxidize the alkanes and partner SRB perform sulfate reduction.

### The enriched archaea activate alkanes with Acrs

Both *Ca.* Alkanophagales MAGs encode three Acrs (*acrABG*) (Extended Data Fig. 3). Currently, only the sister group *Ca.* Syntrophoarchaeum encodes a higher number of Acrs with four copies[14]. The six *acrA* sequences, which code for the catalytic subunit[28], form three clusters of two highly similar sequences, one of each species (≥89% identity) in the *acrA* clade (Fig. 3a and Supplementary Table 6)[12–15]. All clusters are highly similar to *acrA*s of *Ca.* Syntrophoarchaeum (Supplementary Table 6).

Both species highly expressed the *acrA* of the third cluster, placing it among the top 19 (C$_8$) to top 4 (C$_5$) expressed genes (Fig. 3b and Supplementary Table 7). This cluster is phylogenetically closely related to *acrA*s that presumably activate long-chain alkanes, for instance in *Ca.* Methanoliparum[15]. In both MAGs, this *acrA* is spatially separated from the *acrB* and *acrG* subunits (Extended Data Fig. 3), which has been previously reported for *Ca.* Syntrophoarchaeum[14].

As in other Acr-dependent alkane-degrading cultures[14,29], a selective inhibitor of the Mcr/Acr, the CoM analogue

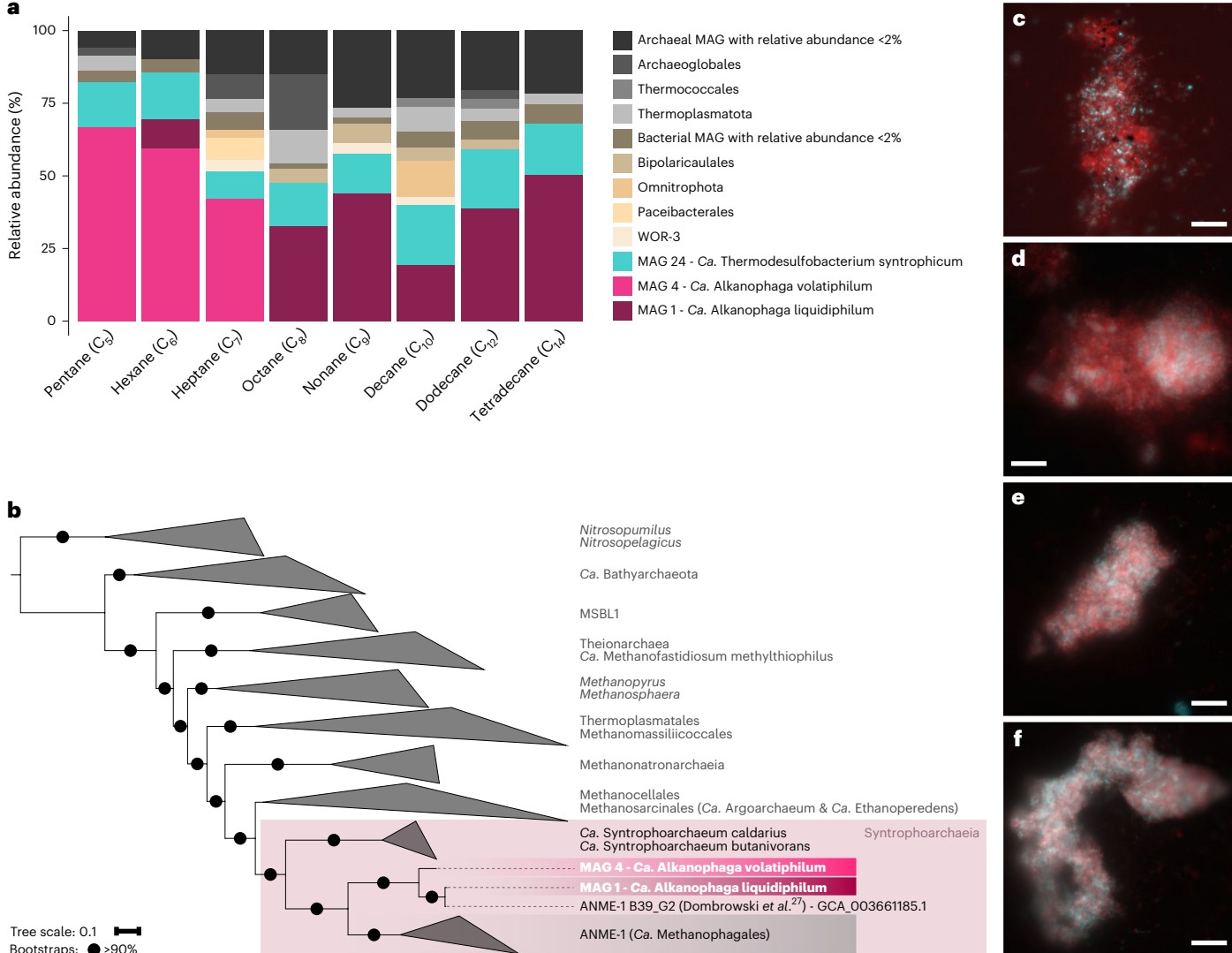

**Fig. 2 | Two archaea of the genus *Ca*. Alkanophaga are abundant in the cultures and closely related to ANME-1. a**, Relative abundances of MAGs obtained from manual binning. *Ca*. Alkanophaga volatiphilum (MAG 4) is abundant in cultures oxidizing shorter, volatile alkanes between $C_5$-$C_7$; *Ca*. Alkanophaga liquidiphilum (MAG 1) is abundant in cultures oxidizing liquid alkanes between $C_8$ and $C_{14}$. A *Thermodesulfobacterium* with the genomic capacities for dissimilatory sulfate reduction, *Ca*. Thermodesulfobacterium syntrophicum, is present in all cultures. Taxonomies of background MAGs are displayed at order level. Background archaea are shaded grey; background bacteria are shaded brown.

**b**, Phylogenomic placement of *Ca*. Alkanophaga MAGs based on the concatenated alignment of 76 archaeal single-copy core genes. *Ca*. Alkanophaga diverge at the root of ANME-1 (*Ca*. Methanophagales). The class Syntrophoarchaeia is highlighted with a shaded rectangle. The outgroup consists of members of the Thermoproteota. Tree scale bar, 10% sequence divergence. **c–f**, Double hybridization of $C_6$ (**c,d**) and $C_{14}$ (**e,f**) culture samples with a specific probe targeting the *Ca*. Alkanophagales clade (Aph183, red) and a general bacterial probe (EUBI-III, cyan). *Ca*. Alkanophaga cells are abundant in the aggregates where they co-occur with bacterial cells. Scale bar, 10 μm.

2-bromoethanosulfonate (BES)[30], suppressed sulfide production (Extended Data Fig. 4a,b), consistent with an Acr-based activation mechanism. Further, metabolite extracts of all cultures contained peaks pertaining to the masses of the corresponding alkyl-CoMs as indicative activation product (Fig. 3c,d and Extended Data Fig. 5). While alkanes from $C_5$-$C_7$ were activated at the first and second carbon atom in similar ratios (Fig. 3c and Extended Data Fig. 5), we observed a shift to more subterminally activated alkanes with increasing alkane length ($\geq C_9$) (Fig. 3d and Extended Data Fig. 5). The longest alkanes $C_{12}$ and $C_{14}$ seemed to be activated predominantly to $\geq$3-alkyl-CoM (Fig. 3d and Extended Data Fig. 5). An activation at multiple positions was previously observed in *Ca*. Syntrophoarchaeum[14]. The comparatively high activation rate at the terminal position for shorter alkanes is unexpected, because particularly in short alkanes, C-H bonds are

stronger at terminal positions compared with subterminal positions[31]. Further degradation of non-terminally activated alkanes probably requires a rearrangement to 1-alkyl-CoM as described for bacterial alkane degradation[32].

We conclude that the archaea represented by MAGs 1 and 4 oxidize the petroleum alkanes. We propose the genus name *Ca*. Alkanophaga, consistent with the previously suggested name *Ca*. Alkanophagales[11], and analogous to the closely related methanotrophs *Ca*. Methanophagales (ANME-1)[33]. The *Ca*. Alkanophaga MAGs share amino acid identities (AAIs) of 55–59% with ANME-1 MAGs (Supplementary Table 8), placing *Ca*. Alkanophaga within the ANME-1 family[34]. On the basis of apparent substrate preference in our enrichment cultures, we propose the names *Ca*. Alkanophaga volatiphilum for the archaeon represented by MAG 4 and *Ca*. Alkanophaga liquidiphilum

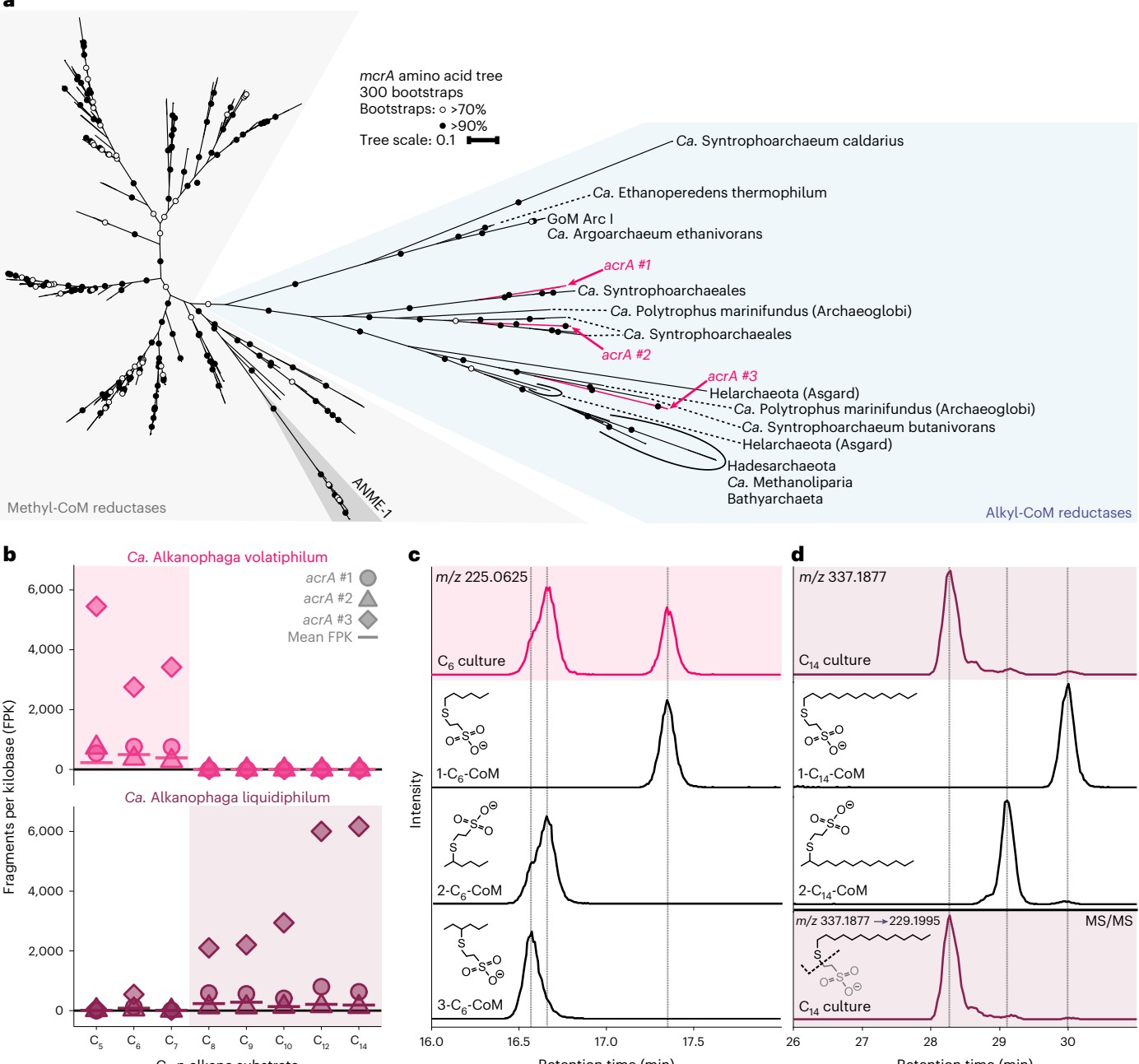

**Fig. 3 | *Ca.* Alkanophaga use alkyl-coenzyme M reductases to activate alkanes to alkyl-CoMs. a**, Phylogenetic placement of translated *mcrA* sequences of *Ca.* Alkanophaga. Both *Ca.* Alkanophaga species contain three *mcrA* sequences, all of which fall into the divergent branch of *mcrA*s, encoding alkyl-CoM reductases (Acrs), highlighted in blue. The six *acrA* sequences form three clusters of two sequences, each cluster containing one sequence of each *Ca.* Alkanophaga species. Tree scale bar, 10% sequence divergence. **b**, Expression of *acrA* genes during growth on various alkanes for both *Ca.* Alkanophaga species. Cultures in which the respective species was prevalent in the metagenomes are highlighted with shaded boxes. The mean expression of all genes of the respective species is shown as a horizontal bar. The *acrA* of the third cluster was strongly expressed, irrespective of substrate length, by the species abundant in that culture.

The expression of the other *acrA* genes was low. **c,d**, Extracted ion chromatograms (EICs) based on exact mass and a window of ±10 mDa of deprotonated ions of variants of $C_6$-CoM (**c**) and $C_{14}$-CoM (**d**) detected via liquid chromatography–mass spectrometry. In both **c** and **d**, the upper shaded panels show the culture extract, with isomers of alkyl-CoM standards below. In **d**, the shaded bottom panel shows the EIC produced with the exact mass of the $C_{14}$-thiolate, a fragmentation product derived in MS/MS experiments from the precursor $C_{14}$-CoM. Dashed vertical lines were added at retention times of peak maxima of standards (**c**) or standards and fragmentation products (**d**) for easier identification of peaks in the culture extracts. While $C_6$ is activated on the first and second carbon atom to a similar degree, $C_{14}$ is activated predominantly to ≥3-$C_{14}$-CoM.

for the archaeon represented by MAG 1. Substrate tests corroborate that *Ca.* A. volatiphilum prefers shorter volatile alkanes <$C_{10}$, while *Ca.* A. liquidiphilum readily degrades all alkanes between $C_6$ and $C_{15}$ (Extended Data Fig. 6).

## *Ca.* Alkanophaga completely oxidize the alkanes to $CO_2$

The oxidation of alkyl-CoMs generated by the Acr requires conversion to acyl-CoA (Fig. 4a,b). The underlying reactions for this transformation are unknown, but for other alkane-degrading archaea, some candidate

enzymes have been proposed. The $C_2$-oxidizing $Ca$. Ethanoperedens thermophilum may catalyse this step with tungstate-containing aldehyde:ferredoxin reductases (Aors). This archaeon encodes three $aor$ copies located closely to genes of the Wood-Ljungdahl (WL) pathway and expresses them during ethane oxidation[13]. While both $Ca$. Alkanophaga encode complete $aor$ gene sets, those genes were only moderately expressed (Supplementary Table 7), casting doubt on a crucial role of the Aor in this reaction in our cultures. A transfer of alkyl moieties to CoA via methyltransferases, as was hypothesized for $Ca$. Syntrophoarchaeum[14], is equally unlikely because of the large alkanes consumed by $Ca$. Alkanophaga. In conclusion, the conversion of alkyl-CoM to acyl-CoA requires further investigation.

Similar to $Ca$. Syntrophoarchaeum[14], $Ca$. Alkanophaga probably processes acyl-CoA to acetyl-CoA units via the β-oxidation pathway[35] (Fig. 4b). $Ca$. Alkanophaga encode all genes for even-chain β-oxidation and expressed them during alkane oxidation (Figs. 4a and 5, Extended Data Fig. 7 and Supplementary Table 7). For odd-chain alkanes, three additional genes are required to degrade the potentially toxic $C_3$-compound propionyl-CoA[36,37], two of which are missing from $Ca$. Alkanophaga. We could not identify complete alternative pathways for the degradation of propionyl-CoA, for example the methylcitrate cycle[37]. Thus, the fate of the propionyl-CoA remains, for the moment, unclear.

Acetyl-CoA units from β-oxidation are shuttled into biomass production or completely oxidized. For the latter, the acetyl-CoA decarbonylase/synthase (ACDS) complex splits a methyl group from acetyl-CoA which is transferred to tetrahydromethanopterin ($H_4$MPT) (Fig. 4b). The enzymes of the $H_4$MPT methyl branch of the WL pathway then oxidize methyl-$H_4$MPT to $CO_2$[13,14]. Both $Ca$. Alkanophaga species encode and expressed multiple ACDS and all enzymes of the WL pathway, except methylene-$H_4$MPT-deyhdrogenase ($mtd$) missing in $Ca$. A. volatiphilum (Figs. 4a and 5, Extended Data Fig. 7 and Supplementary Table 7).

Unlike the closely related $Ca$. Syntrophoarchaeum and ANME-1, both $Ca$. Alkanophaga encode several 5,10-methylene-$H_4$MPT reductase ($mer$) genes. This enzyme catalyses the oxidation of methyl-$H_4$MPT ($CH_3$-$H_4$MPT) to methylene-$H_4$MPT ($CH_2$ = $H_4$MPT) in the first step of the oxidative WL pathway[38]. Two of these genes, OD814_001315 in $Ca$. A. volatiphilum and OD815_000385 in $Ca$. A. liquidiphilum, most probably code for a canonical $mer$ because they are highly similar (>99%) to $mer$ copies of Methanomicrobia. A phylogenetic analysis placed these two $mer$ sequences next to each other and close to those of the hydrogenotrophic methanogens Methanocellales[39] (Extended Data Fig. 8a). We therefore hypothesize that $Ca$. Alkanophaga inherited $mer$ vertically from the methanogenic ancestor of Methanocellales. $Ca$. Syntrophoarchaeum and ANME-1 seem to have replaced $mer$ with methylene-tetrahydrofolate ($H_4$F) reductase ($metF$) of the $H_4$F methyl branch of the WL pathway[14,40]. Both $Ca$. Alkanophaga MAGs also encode $metF$ copies, which are highly similar (70−80%) to those of $Ca$. Syntrophoarchaeum and cluster next to $metF$ sequences of Hadarchaeota from Jinze hot spring (China) and Yellowstone National Park (USA) (Extended Data Fig. 8b). While both $mer$ and $metF$ were transcribed, $mer$ was especially expressed by $Ca$. A. liquidiphilum in cultures oxidizing longer alkanes ≥$C_{10}$ (Fig. 5b,d, Extended Data Fig. 7e,f,k,l and Supplementary Table 7).

### $Ca$. Alkanophaga partner with a $Thermodesulfobacterium$

$Ca$. Alkanophaga lack the dissimilatory sulfate reduction (DSR) pathway and therefore require a partner organism. We identified a $Thermodesulfobacterium$ represented by MAG 24, which was enriched in all cultures (Fig. 2a and Supplementary Table 4) and rare in the original slurry, as the most likely syntrophic sulfate reducer. MAG 24 encodes and expressed the three DSR proteins ATP-sulfurylase (Sat), APS-reductase (Apr) and dissimilatory sulfite reductase (Dsr)[41] (Fig. 5, Extended Data Fig. 7 and Supplementary Table 7). We propose the name $Ca$. Thermodesulfobacterium syntrophicum for this bacterium, which is closely related to the hyperthermophilic sulfate-reducing $Thermodesulfobacterium geofontis$ isolated from the Obsidian Pool in Yellowstone National Park (USA)[42] (Extended Data Fig. 9).

**Etymology.** Alkanophaga: $alkano$ (new Latin): alkane and $phaga$ (Greek): eating; volatiphilum: $volatilis$ (Latin): volatile and $philum$ (Greek): preferring; liquidiphilum: $liquidus$ (Latin): liquid and $philum$ (Greek): preferring; syntrophicum: $syn$ (Greek): together with; $trephein$ (Greek): nourish and $icum$ (Latin): pertaining to.

**Locality.** Hydrothermally heated oil-rich deep-sea sediment in the Guaymas Basin, Gulf of California, Mexico.

**Description.** $Ca$. Alkanophaga volatiphilum and $Ca$. Alkanophaga liquidiphilum: thermophilic, anaerobic, petroleum ($C_5$-$C_{14}$) $n$-alkane-oxidizing archaea, forming syntrophic consortia with the sulfate-reducing $Ca$. Thermodesulfobacterium syntrophicum.

Syntrophic microorganisms trade electrons via molecular intermediates, such as hydrogen or formate[43], or direct interspecies electron transfer (DIET)[44]. Both $Ca$. Alkanophaga and $Ca$. T. syntrophicum encode membrane-bound [NiFe]-hydrogenases, including several hydrogenase maturation factors, enabling electron transfer via molecular hydrogen. Some hydrogenase genes were substantially expressed (Fig. 5, Extended Data Fig. 7 and Supplementary Table 7). Formate dehydrogenases, necessary for electron transfer via formate, were also present in both partners and moderately expressed (Fig. 5, Extended Data Fig. 7 and Supplementary Table 7). However, the addition of hydrogen or formate did not accelerate sulfide production (Extended Data Fig. 4c,d). Moreover, cultures in which sulfate reduction was inhibited by the addition of sodium molybdate produced only miniscule fractions (max. 2.4% for $C_6$ and 0.9% for $C_{14}$) of the hydrogen concentrations that would be necessary were hydrogen the sole electron carrier (Supplementary Table 9). Thus, neither molecular hydrogen nor formate are probably primary electron carriers.

Alternatively, alkane oxidation and sulfate reduction are coupled through DIET, as suggested for other alkane-oxidizing consortia[13,14,45]. DIET probably involves cell appendages, such as bacterial type IV pilin (PilA) or the archaeal flagellin B (FlaB), and multihaem $c$-type cytochromes (MHCs), forming conductive nanowires enabling electron transport[46,47]. Both components are present and strongly expressed in previously established alkane-oxidizing consortia[13,14,45]. Surprisingly, neither our nor the previously published $Ca$. Alkanophaga MAGs encode any MHCs, while the closest relatives of $Ca$. Alkanophaga, ANME-1 and $Ca$. Syntrophoarchaeum, encode multiple MHCs[14,33]. $Ca$. T. syntrophicum encodes six MHCs, only one of which was slightly enriched in all cultures (Supplementary Table 7). This implies a minor role of MHCs in the interaction of both organisms.

Both $Ca$. Alkanophaga encode several copies of $pilA$ and $flaB$ for the formation of cell appendages for DIET. These genes were among the most highly expressed genes of $Ca$. Alkanophaga in all cultures. $Ca$. T. syntrophicum encodes several $pilA$ genes as well, some of which were strongly enriched in the $C_{10}$-$C_{14}$ cultures (Supplementary Table 7). Transmission electron microscopy revealed diffuse filamentous structures in the intercellular space that might pertain to such nanowires (Extended Data Fig. 10), but further analyses are necessary to confirm the identity of these structures.

We predict that electron transfer in our cultures is based predominantly on DIET. The lack of MHCs in $Ca$. Alkanophaga might be compensated by MHC production in the partner bacterium similar to observations in syntrophic methane-oxidizing cultures, where only the bacterial partner expressed $pilA$ genes[45]. Alternatively, DIET might be completely independent of MHCs, which has been observed before[48,49]. It remains possible that a small fraction of electrons are transferred via soluble intermediates such as hydrogen. Such a combination of

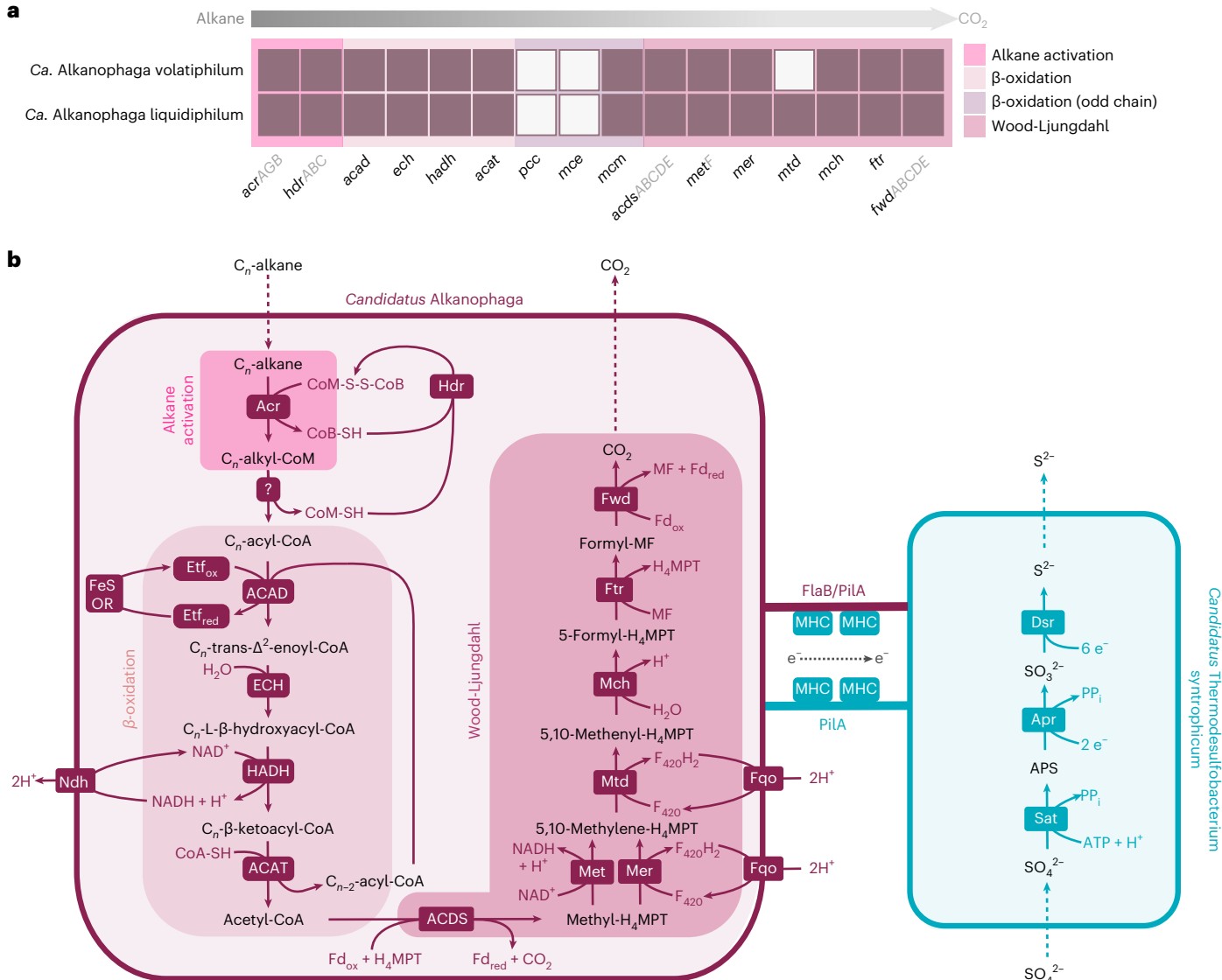

**Fig. 4 | Mechanism of syntrophic petroleum alkane oxidation. a**, Genomic capacities for alkane oxidation in *Ca*. Alkanophaga MAGs. Colour-filled rectangles indicate presence of a gene; white rectangles indicate absence. For multiple-subunit proteins, at least one gene coding for each subunit was found in case of a filled rectangle. **b**, Metabolic model for syntrophic alkane oxidation. *Ca*. Alkanophaga activates alkanes via the alkyl-coenzyme M reductase (Acr). A yet unknown pathway transforms alkyl-CoM to acyl-CoA. The enzymes of the β-oxidation pathway, including (1) acyl-CoA dehydrogenase (ACAD), (2) enoyl-CoA hydratase (ECH), (3) hydroxyacyl-CoA dehydrogenase (HADH) and (4) acyl-CoA acetyltransferase (ACAT), cleave acyl-CoA into multiple acetyl-CoA units. The acetyl-CoA decarbonylase/synthase (ACDS) complex breaks the acetyl units into $CO_2$ and a tetrahydromethanopterin ($H_4$MPT)-bound methyl unit. The methyl branch of the Wood-Ljungdahl pathway, including (1) 5,10-methylene tetrahydrofolate reductase (MetF) and/or 5,10-methylene $H_4$MPT reductase (Mer), (2) methylene-$H_4$MPT dehydrogenase (Mtd), (3) methenyl-$H_4$MPT

cyclohydrolase (Mch), (4) formylmethanofuran-$H_4$MPT formyltransferase (Ftr) and (5) tungsten-containing formylmethanofuran dehydrogenase (Fwd), oxidizes methyl-$H_4$MPT to $CO_2$. Most probably, an electron transfer flavoprotein (Etf) serves as electron acceptor in the first step of the β-oxidation pathway. Cofactor recycling is taken over by cytoplasmic heterodisulfide reductase (Hdr), [FeS]-oxidoreductase (FeS-OR), NADH dehydrogenase (Ndh) and $F_{420}H_2$:quinone oxidoreductase (Fqo). Electrons from alkane oxidation are transferred to *Ca*. Thermodesulfobacterium syntrophicum, most probably via DIET. DIET seems to rely on conductive filaments formed by type IV pilin (PilA) and/or flagellin B (FlaB) that are expressed by both partners, and multihaem *c*-type cytochromes (MHCs) expressed solely by the bacterium. Sulfate reduction in *Ca*. T. syntrophicum follows the canonical dissimilatory sulfate pathway using the enzymes ATP-sulfurylase (Sat), APS-reductase (Apr) and dissimilatory sulfite reductase (Dsr). *pcc*, gene encoding propionyl-CoA decarboxylase; *mce*, gene encoding methylmalonyl-CoA epimerase.

DIET with diffusion-based electron transport was recently shown to be energetically favourable for syntrophic consortia[50].

## Discussion

Petroleum-rich anoxic environments such as oil reservoirs, oily sludges and polluted sediments harbour oil-degrading microorganisms. Isolates from these environments that couple petroleum alkane oxidation

to sulfate reduction are mostly bacteria active at temperatures ≤60 °C (ref. 51). With *Ca*. Alkanophaga, we enriched a thermophilic clade thriving on petroleum alkanes from $C_5$ to $C_{14}$ at temperatures between 65–75 °C (Extended Data Fig. 4e,f), which approach the suggested upper limit of microbial hydrocarbon degradation in petroleum reservoirs of around 80 °C (ref. 52). This temperature optimum is reflected by the high relative abundance of *Ca*. Alkanophaga in deep, heated sediment

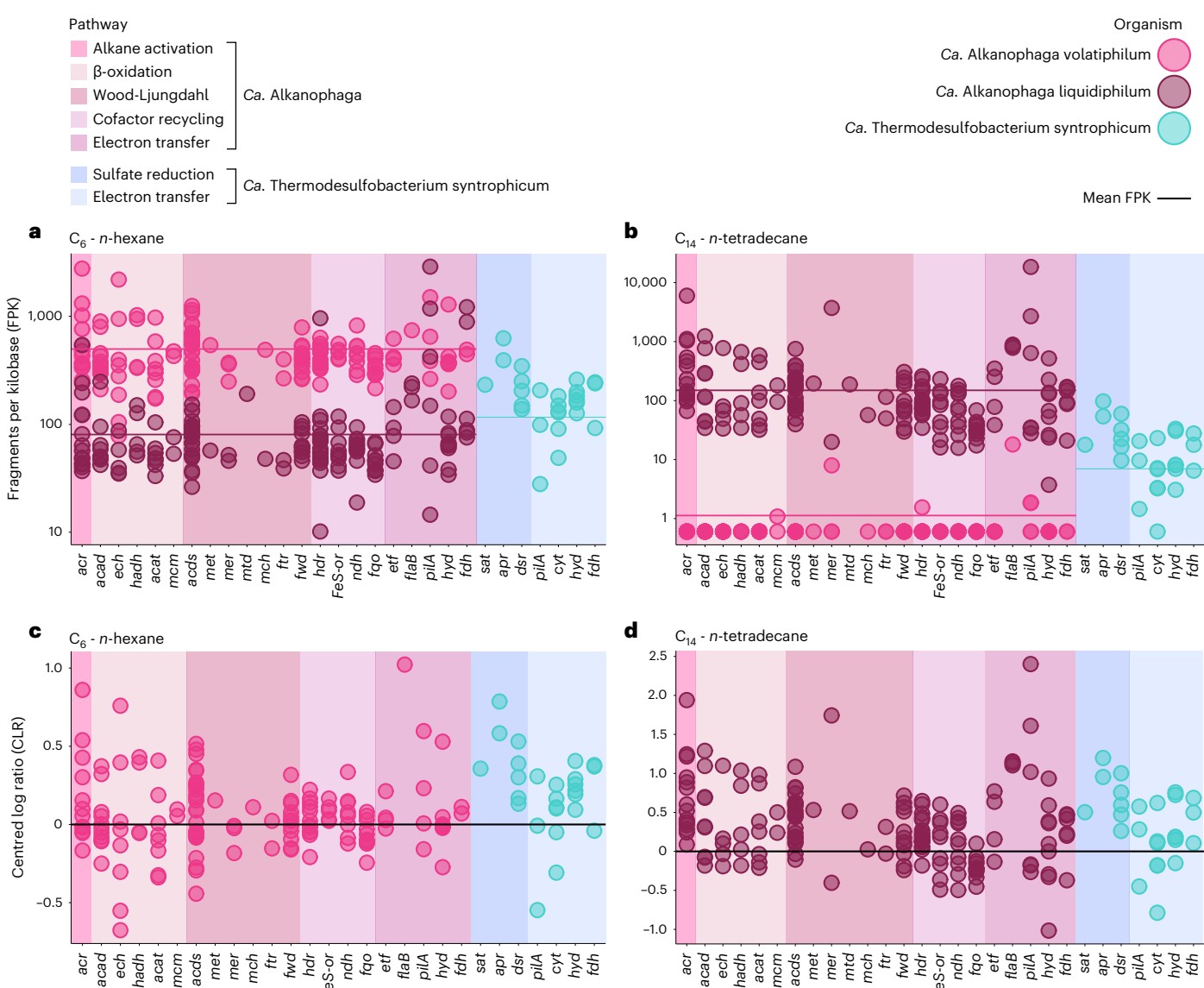

**Fig. 5 | Gene expression profiles for syntrophic petroleum alkane oxidation.** **a,b**, Fragment counts normalized to gene length (FPK) shown on a logarithmic *y* axis. The average gene expression of each organism is indicated as arithmetic mean (sum of all FPK values divided by number of genes) depicted as a horizontal line. **c,d**, Fragment counts normalized as CLR. For simplicity, only the values of the more active *Ca.* Alkanophaga species are shown. For abbreviations, see Fig. 4; *hyd*, gene encoding [NiFe]-hydrogenase; *fdh*, gene encoding formate dehydrogenase; *cyt*, gene encoding multihaem cytochrome.

layers of the Guaymas Basin, inferring a crucial role of these archaea in thermophilic hydrocarbon transformation.

*Ca.* Alkanophaga encode three Acrs for anaerobic alkane activation, one less than the closely related short-chain alkane oxidizer *Ca.* Syntrophoarchaeum[14]. Independent of alkane length, *Ca.* Alkanophaga strongly expressed only one of the Acrs, which is highly similar to the highest expressed Acr in *Ca.* Syntrophoarchaeum during $C_4$ oxidation[14]. Future studies may reveal functions or substrates of the other two lower expressed Acrs. *Ca.* Alkanophaga stand out among Acr-using archaea with their wide substrate range between $C_5$ and $C_{15}$. Therewith, all alkanes between $C_1$ and $C_{20}$ are confirmed substrates of alkane-oxidizing archaea[12–15]. Our study implies that substrate flexibility of the Acr increases with increasing alkane length, which is presumably enabled by a wider catalytic cleft in the Acrs activating $C_3+$ alkanes[31] compared with the highly selective hydrophobic tunnel detected in the $C_2$-activating Acr[29]. Crystallization efforts may resolve molecular and structural modifications of these Acrs that make use of such a wide substrate spectrum.

The three clades of the class Syntrophoarchaeia (*Ca.* Alkanophaga, *Ca.* Syntrophoarchaeum and ANME-1), share many metabolic features such as obligate syntrophic growth with partner SRB and presence of the β-oxidation and WL pathways. At the same time, they exhibit remarkable metabolic and genomic differences. For instance, ANME-1 encode the canonical Mcr for methane metabolism, which is missing in *Ca.* Syntrophoarchaeum and *Ca.* Alkanophaga, preventing them from oxidizing and producing methane. Instead, the latter two possess multiple multicarbon alkane-activating Acrs, which are in turn absent in ANME-1. Our study supports the previously established hypothesis that multicarbon alkane metabolism probably preceded methanotrophy in the Syntrophoarchaeia[11,53] because of the basal position of both multicarbon alkane oxidizers (Fig. 2b) and their similar metabolisms. The presence of the β-oxidation pathway in ANME-1 (ref. 11) supports this notion because this pathway is required for the oxidation of $C_3+$ alkanes but serves no purpose in the oxidation of methane. We propose that the common ancestor of the Syntrophoarchaeia was a multicarbon alkane-oxidizing archaeon with multiple Acrs. *Ca.* Syntrophoarchaeum

and *Ca*. Alkanophaga emerged from this ancestor, preserving a similar metabolism. Today, *Ca*. Syntrophoarchaeum thrives at much lower temperatures (50 °C) and seems incapable of oxidizing liquid alkanes[14]. Thus, adaptation to different temperatures and substrates might have enabled *Ca*. Syntrophoarchaeum and *Ca*. Alkanophaga to occupy different ecological niches. *Ca*. Alkanophaga and ANME-1 also shared a common ancestor from which ANME-1 probably diverged after losing their Acrs[54] and acquiring an Mcr, potentially from a methanogen via lateral gene transfer[33,55].

*Ca*. Alkanophaga differ from the two other groups of the Syntrophoarchaeia in two main aspects. First, *Ca*. Alkanophaga encode and expressed *mer*, an essential enzyme of the canonical methanogenesis pathway[56]. ANME-1, except for a putative methanogenic ANME-1 member[57], and *Ca*. Syntrophoarchaeum lack *mer* and instead code for the phylogenetically widely distributed *metF*[14,33,58], which is also present and expressed in *Ca*. Alkanophaga. We hypothesize that *mer* in *Ca*. Alkanophaga is a remnant from a methanogenic ancestor. Second, *Ca*. Alkanophaga lack MHCs, which are often considered essential for DIET between syntrophic partners[46]. All other syntrophic alkane-oxidizing archaea code for several MHCs[53]. However, an absence of MHCs in DIET-performing methanogens has been recognized before[48]. It is thus conceivable that MHCs aid in but are not essential for DIET and that MHCs were potentially lost by *Ca*. Alkanophaga without a substantial impact on the efficiency of electron transfer. The loss of all MHCs opens up questions as to the mechanisms that occurred. In a recent study, giant extrachromosomal elements named Borgs, many of which carried clusters of MHCs, were reconstructed from methane-oxidizing *Methanoperedens* (ANME-2d) archaea[59]. One could imagine that MHCs in the Syntrophoarchaeia ancestor were encoded on such a Borg, which was then lost by *Ca*. Alkanophaga. This could explain why all MHCs are absent in *Ca*. Alkanophaga. However, the presence of Borgs in other members of the Syntrophoarchaeia still needs to be examined.

*Ca*. Alkanophaga partner with the sulfate-reducing *Ca*. Thermodesulfobacterium syntrophicum. Previously enriched alkane-oxidizing archaea partner with a different bacterium, *Ca*. Desulfofervidus auxilii, which has an optimal growth temperature of 60 °C (refs. [13,14,16,17]). We suspect that the higher incubation temperature of our study selected for a more thermophilic partner organism. Recently, another *Thermodesulfobacterium* species, *Ca*. Thermodesulfobacterium torris (ANI 84.0%, Extended Data Fig. 9), has been reported as syntrophic sulfate reducer partnering with thermophilic ANME-1c at 70 °C (ref. [60]). Thus, Thermodesulfobacteria represent a new group of partner organisms for alkane-oxidizing archaea at high temperatures. In contrast to *Ca*. Alkanophaga, *Ca*. T. syntrophicum encodes and expressed several MHCs, which could support DIET for both partners.

All currently available *Ca*. Alkanophaga sequences originate from the Guaymas Basin, a thoroughly studied hydrothermal vent area hauling heated fluids rich in alkanes[61]. We suspect two main reasons for this apparent absence in other environments. First, until recently, microbial community studies have mostly focused on 16S ribosomal (r)RNA gene amplification and sequencing, a method depending heavily on primer choice[62]. We discovered a mismatch of the commonly used archaeal primer Arch915 (5′-GTGCTCCCCCGCCAATTC**C**T-3′[63], mismatch in bold) to the 16S rRNA gene sequences of *Ca*. Alkanophaga, which probably produces an artificial underrepresentation of *Ca*. Alkanophaga in public databases. Second, sequencing data from other environments similar to the Guaymas Basin, that is, heated oil reservoirs with sulfate supply, remains scarce. Many of these reservoirs, often buried kilometres deep within the subsurface, are extremely hard to access[64]. In addition, the risk of contamination from the upper biosphere during sampling increases with depth, which might conceal the native community[64]. Still, sampling technologies have greatly improved in recent years, and the focus has shifted from amplification-based 16S rRNA gene to shotgun metagenome studies, which should facilitate a more accurate molecular characterization of reservoir microorganisms.

Thus, future studies may disclose the coexistence and activity of *Ca*. Alkanophaga and *Ca*. T. syntrophicum in other heated, petroleum-rich environments.

## Methods

All chemicals were of analytical grade and obtained from Sigma Aldrich, unless otherwise stated. All incubations were done under gentle shaking (40 r.p.m.) in the dark.

### Cultivation of anaerobic thermophilic alkane degraders

The push core used for anoxic cultivations was collected with submersible *Alvin* during RV *Atlantis* cruise AT42-05 in the Guaymas Basin (Gulf of California, Mexico) (dive 4,991, core 15, 27° 00′ 41.1″ N, 111° 24′ 16.3″ W, 2,013 m water depth, 17 November 2018). While shipboard, the push core was transferred to a sealed glass bottle, purged with argon and stored at 4 °C. In the home laboratory, an anoxic sediment slurry was prepared with synthetic sulfate-reducer medium (SRM)[65], using a ratio of 10% sediment and 90% SRM (v/v), and distributed in 100 ml portions into culture bottles. Cultures were supplemented with 200 µl liquid alkane ($C_5$-$C_{14}$) in duplicates. For the $C_5$-$C_{10}$ alkanes, 4 ml 2,2,4,4,6,8,8-heptamethylnonane (HMN) were added to mitigate potential toxic effects of the substrate[66]. A substrate-free culture served as a negative control. Headspaces were filled with $N_2$:$CO_2$ (90:10; 1 atm overpressure) and incubated at 70 °C.

Sulfide production was measured every 2–4 weeks using a copper sulfate assay[67]. Once sulfide concentrations reached 12–15 mM, cultures were diluted 1:3 with SRM and supplied with fresh substrate. Activity doubling times were determined from the development of sulfide concentrations during the first two dilutions. Sulfide concentrations over time were displayed using a logarithmic (base 2) $y$ axis. An exponential trend line with the formula $y = n \times e^{mx}$ was generated. Per definition, the doubling time equals $\frac{\ln(2)}{m}$.

### Quantitative substrate turnover experiment

Triplicate 100 ml dilutions with 20 ml headspace were prepared from $C_6$- and $C_{14}$-oxidizing cultures, supplied with substrate and incubated at 70 °C, complemented by a substrate-free negative control. Sulfate and DIC concentrations were measured from weekly subsamples until the cultures had reached sulfide concentrations of ≥15 mM. Samples were sterile filtered using a GTTP polycarbonate filter (0.2 µM pore size; Millipore). For DIC measurements, 1 ml filtrate was transferred into synthetic-air-purged 12 ml Exetainer vials (Labco) filled with 100 µl phosphoric acid (45%). After 10 h of equilibration, headspace DIC was measured by isotope ratio infrared spectroscopy (Thermo Fisher; Delta Ray IRIS with URI connect and Cetac ASX-7100 autosampler) with standards of known concentration. To determine sulfate concentrations, 1 ml of the filtrate was fixed in 0.5 ml 100 mM zinc acetate. The sample was centrifuged and the clear supernatant was diluted 1:50 in deionized water. Sulfate was measured by ion chromatography (930 compact IC, Metrohm) against standards with known concentrations.

### DNA extraction and short-read sequencing

DNA was extracted from pellets of 25 ml culture samples collected after the third dilution, using a modified SDS-based extraction method as previously described[68]. Total DNA yield per sample, determined by fluorometric DNA concentration measurement, ranged from 0.9 µg to 3.6 µg. Samples were sequenced at the Max Planck-Genome-Centre (Cologne, Germany). $C_6$-$C_{14}$ culture samples were sequenced as 2 × 250 paired-end reads on an Illumina HiSeq2500 sequencing platform. The $C_5$ culture sample was sequenced later because of slower growth, together with a sample of the sediment slurry before incubation, by which time the sequencing facility had changed their settings to 2 × 150 bp paired-end reads on an Illumina HiSeq3000 platform. Between 4,140,953 and 4,234,808 raw reads were obtained per culture sample. From the original slurry, 3,130,329 reads were gained.

## Short-read DNA data analysis

Reads from short-read metagenome sequencing were quality-trimmed using BBDuk (included in BBMap v.38.79; https://sourceforge.net/projects/bbmap/; minimum quality value: 20, minimum read length: 50). Reads of the $C_6$-$C_{14}$ samples were coassembled using SPAdes (v.3.14.0; https://github.com/ablab/spades)[69], running BayesHammer error correction and $k$-mer increments (21, 33, 55, 77, 99 and 121) with default settings. The output scaffolds were reformatted using anvi'o (v.7; https://github.com/merenlab/anvio/releases/)[70], simplifying names and removing contigs shorter than 3,000 bps. Trimmed reads were mapped back to the reformatted scaffolds using Bowtie2 (v.2.3.2; http://bowtie-bio.sourceforge.net/bowtie2/index.shtml)[71] in the local read alignment setting. Sequence alignment map files were converted to binary alignment map (BAM) files with SAMtools (v.1.5; http://samtools.sourceforge.net/)[72] and indexed with anvi'o. A contigs database was created from the reformatted scaffolds and profile databases were generated for each sample with anvi'o. Profile databases were merged, enforcing hierarchical clustering. Hidden Markov model (HMM) searches were run via anvi'o on the contigs database to detect genes encoding for Mcrs/Acrs, Wood-Ljungdahl pathway and DSR. Taxonomies for open reading frames were imported into the contigs database using the Centrifuge classifier (v.1.0.2-beta; https://ccb.jhu.edu/software/centrifuge/)[73]. The contigs database was inspected in the anvi'o interactive interface, which clusters the contigs hierarchically on the basis of sequence composition and differential coverage, thereby indicating their relatedness to each other[70]. Binning was performed manually in the interface by clicking branches of the dendrogram in the centre of the interface and using the GC content, mean coverage in the samples, gene taxonomy and real-time statistics on completion and redundancy based on single-copy core genes as guides. The dendrogram branches were followed systematically in a counterclockwise direction to obtain the maximum number of bins. Bin quality was assessed again with CheckM (v.1.1.3; https://ecogenomics.github.io/CheckM/)[74] and only bins with completeness >50% and redundancy <10% were kept. Taxonomies were assigned to these metagenome-assembled genomes (MAGs) using GTDB-Tk (v.1.5.1; https://github.com/Ecogenomics/GTDBTk)[75]. All manually generated MAGs were refined with anvi'o to minimize contamination. We identified MAGs 1 and 4 as the likely alkane oxidizers and MAG 24 as the likely sulfate reducer based on their mean coverages and HMM hits. To increase the completeness of these three MAGs, an iterative reassembly loop (https://github.com/zehanna/MCA70_analysis/targeted_reassembly_loop.sh) was performed. Therein, the trimmed reads were repeatedly mapped to the refined MAG using BBMap with a minimum alignment identity of 97%. Mapped reads were then assembled using SPAdes. The assembly was quality-checked with CheckM and used as a new reference file to map the trimmed reads to. After performing 25 iterations of this loop, the assembly with the highest quality (that is, highest completeness, lowest contamination and lowest strain heterogeneity) was selected for further analysis. Final MAGs were annotated with Prokka (v.1.14.6; https://github.com/tseemann/prokka)[76] and the anvi'o-integrated databases NCBI clusters of orthologous genes (COGs)[77], Kyoto Encyclopedia of Genes and Genomes (KEGG)[78], Protein Families (Pfams)[79] and KEGG orthologues HMMs (KOfams)[80]. A bash script (https://github.com/zehanna/MCA70_analysis/CxxCH_scan.sh) was run to search for the haem-binding CxxCH amino acid motif[81] in the translated gene sequences of the three MAGs. Selected translated gene sequences were exported for gene calls from the contigs database with anvi'o and compared via the BLASTp[82] web interface (http://www.ncbi.nlm.nih.gov/blast).

Relative abundances of the MAGs were calculated by mapping the trimmed reads to the manually curated and refined MAGs with CoverM (v.0.6.1; https://github.com/wwood/CoverM) in genome mode including the dereplication flag using the default aligner Minimap2 (v.2.21; https://docs.csc.fi/apps/minimap2/) in short-read mode, discarding unmapped reads. The final relative abundance of each MAG is the percentage of the MAG in the mapped fraction of each sample. ANIs between MAGs were calculated with FastANI (v.1.32; https://github.com/ParBLiSS/FastANI).

Because of later sequencing, the original slurry and $C_5$ samples were treated separately from the previously sequenced samples and assembled individually. We could not obtain quality MAGs for the original slurry sample; therefore, we estimated the phylogenetic composition on the basis of reconstructed small subunit ribosomal RNAs (SSU rRNAs) mapped against the SILVA SSU reference database (v.138.1)[83] with phyloFlash (v.3.4.1; https://github.com/HRGV/phyloFlash)[84]. For the $C_5$ sample, the same procedure as for the previously sequenced culture samples was followed. The identity (ANI ≥ 95%; ref. [85]) of the *Ca.* Alkanophaga volatiphilum and *Ca.* Thermodesulfobacterium syntrophicum MAGs from the $C_5$ sample, MAG 4_1 and MAG 24_1, respectively, to the previously reconstructed ones was confirmed via FastANI.

To estimate relative abundances of *Ca.* Alkanophaga and *Ca.* T. syntrophicum MAGs in the original slurry, the trimmed reads of the original slurry were mapped to the MAGs with CoverM.

## Construction of phylogenomic trees for archaea and bacteria

The archaeal tree was constructed using 98 publicly available Halobacteriota and Thermoproteota genomes (Supplementary Table 10) from NCBI plus the *Ca.* Alkanophaga MAGs from this study. For the bacterial tree, 121 publicly available Desulfobacterota and Bipolaricaulota genomes (Supplementary Table 10) and the *Thermodesulfobacterium* MAG from this study were included. Trees were based on the concatenated alignment of 76 single-copy core genes (SCG) for archaea and 71 SCGs for bacteria. Alignments were generated with anvi'o, which uses the multiple sequence alignment tool MUSCLE[86] (v.5.1; https://github.com/rcedgar/muscle). Trees were calculated with RAxML (randomized accelerated maximum likelihood) (v.8.2.12; https://cme.h-its.org/exelixis/web/software/raxml/)[87] using the PROTGAMMAAUTO model and autoMRE option, which required 50 iterations to reach a convergent tree for both alignments. Trees were visualized with the Interactive Tree of Life online tool (https://itol.embl.de/)[88]. To resolve taxonomic levels, the *Ca.* Alkanophaga MAGs were compared to the ANME-1 and *Ca.* Syntrophoarchaeales MAGs included in the tree by calculating average amino acid identities (AAIs) using the aai_wf feature of the CompareM software (v.0.1.2; https://github.com/dparks1134/CompareM) with default settings.

## In situ hybridization and microscopy

Culture samples were fixed in 1% formaldehyde for 1 h at r.t., washed twice in 1× PBS and stored in 1× PBS-ethanol (1:1 v/v) at −20 °C. Aliquots were filtered onto GTTP polycarbonate filters (0.2 µM pore size; Millipore). Filters were embedded in 0.2% agarose. For permeabilization, three consecutive treatments were performed: (1) lysozyme solution (0.05 M EDTA (pH 8.0), 0.1 M Tris-HCl (pH 7.5) and 10 mg ml$^{-1}$ lysozyme in MilliQ-grade deionized water) for 1 h at 37 °C; (2) proteinase K solution (0.05 M EDTA (pH 8.0), 0.1 M Tris-HCl (pH 7.5) and 7.5 µg ml$^{-1}$ proteinase K in MilliQ) for 10 min at r.t.; and (3) 0.1 M HCl solution for 5 min at r.t. Endogenous peroxidases were inactivated using 0.15% $H_2O_2$ in methanol for 30 min at r.t. A specific probe was designed to exclusively target the *Ca.* Alkanophagales clade. Therefore, the *Ca.* Alkanophaga 16S rRNA gene sequences were added to the SILVA SSU reference database (v.138.1) using the ARB software[89] (v.7.1; http://www.arb-home.de/home.html). A subtree containing all ANME-1 16S rRNA gene sequences, plus the two sequences from *Ca.* Alkanophaga, was calculated using RAxML (v.8; https://cme.h-its.org/exelixis/web/software/raxml/) with 100 bootstrap replicates, a 50% similarity filter, the GTRGAMMA model and *Methanocella* as outgroup. The probe was generated using the probe design feature with these parameters: length of probe, 19 nucleotides; temperature, 50–100 °C; GC content, 50–100%; *E. coli* position, any; max. non-group hits, 5; min. group hits, 100%. Criteria for candidate probes were: GC content lower than

60%, lowest possible number of matches to non-group species with decreasing temperature, at least one mismatch to non-group species. We ordered a probe that fit these criteria (Aph183) with the sequence 5′-GCATTCCAGCACTCCATGG-3′ from Biomers. For bacteria, the general probe combination EUBI-III (I: 5-GCTGCCTCCCGTAGGAGT-3; II: 5-GCAGCCACCCGTAGGTGT-3; III: 5-GCTGCCACCCGTAGGTGT-3)[90] was applied. Probe working solution (50 ng µl⁻¹) was diluted 1:300 in hybridization buffer containing 30% formamide for Aph183 and 35% formamide for EUBI-III. Probes were hybridized at 46 °C for 3–4 h. Signals were amplified with tyramides labelled with Alexa Fluor 488 for bacteria and Alexa Fluor 594 for *Ca*. Alkanophaga (Thermo Fisher) for 45 min at 46 °C. For double hybridizations, peroxidases from the first hybridization were inactivated using 0.30% $H_2O_2$ in methanol for 30 min at r.t. before the second hybridization and amplification. Filters were analysed via epifluorescence microscopy (Axiophot II imaging; Zeiss). Images were captured with the AxioCamMR camera and the AxioVision software included in the microscope. Images were processed using ImageJ (v.1.49, https://imagej.nih.gov/ij/), where the colour of Alexa488 was changed to cyan to improve accessibility.

### Phylogenetic analysis of proteins involved in alkane oxidation in *Ca*. Alkanophaga

For the *mcrA* tree, the six full-length *mcrA* sequences of *Ca*. Alkanophaga were aligned with 347 publicly available *mcrA* sequences. For the *mer* and the *metF* trees, *Ca*. Alkanophaga sequences were added to publicly available alignments in ref. 33 (*mer*: Fig04B; *metF*: Fig05C of Supplement S1). Sequences were aligned with MAFFT (multiple alignment using fast Fourier transform) (v.7.475; https://mafft.cbrc.jp/alignment/software/)[91]. Alignments were trimmed with SeaView (v.5; http://doua.prabi.fr/software/seaview)[92]. For the *mcrA* tree, sequences shorter than 450 amino acids were removed after trimming, after which 337 sequences remained (Supplementary Table 10). Trees were calculated with RAxML (v.8.2.4) using the PROTGAMMAAUTO model, which assigned LG with empirical base frequencies as amino acid model and the autoMRE option for bootstraps, which required 300, 550 and 400 iterations to reach a consensus tree for the *mcrA*, *mer* and *metF* alignments, respectively. Trees were visualized with the Interactive Tree of Life online tool (https://itol.embl.de/)[88].

### RNA extraction and short-read sequencing

For total RNA extraction, 10 ml of culture material collected after the third dilution at the exponential growth stage were filtered through an RNAse-free cellulose nitrate filter (pore size 0.45 µm; Sartorius). Immediately after filtration, filters were incubated with 5 ml RNAlater for 30 min. RNA was extracted from filters using the Quick-RNA miniprep kit (Zymo Research). DNA was digested without RNase inhibitor. No rRNA depletion step was performed. Between 0.3 and 1.3 µg of total RNA were obtained per sample as determined by fluorometric RNA concentration measurement. Samples were sequenced as 2 × 250 (C₅: 2 × 150) paired-end reads at the Max Planck-Genome-Centre on the Illumina HiSeq2500 (C₅: Illumina HiSeq3000) sequencing platform. Between 4,043,349 and 4,785,231 raw reads were obtained per sample.

### Short-read RNA data analysis

Reads from metatranscriptome sequencing were quality-trimmed using BBDuk (included in BBMap v.38.79). Trimmed reads were mapped to the concatenated *Ca*. Alkanophaga MAGs to minimize unspecific mapping because of the high similarity of the two MAGs and to the *Ca*. Thermodesulfobacterium syntrophicum MAG using BBMap (v.38.87) with minimal alignment identity of 98%. Mapped reads were counted using featureCounts (v.1.4.6-p5; http://subread.sourceforge.net/)[93] with minimum required number of overlapping bases and minimum mapping quality score of 10, counting fragments instead of reads.

Before normalization, rRNA reads were excluded. Fragments were first normalized to gene length, yielding fragments per kilobase (FPK).

$$FPK_i = \frac{C_i}{L_i} \quad (2)$$

The centred-log ratio (CLR) was calculated as the base-10 logarithm of read count $C_i$ of gene $i$ normalized by gene length $L_i$ in kilobases and divided by the geometric mean of all read counts $C_1 - C_n$ normalized by their respective gene length $L_1 - L_n$.

$$CLR_i = \log\left( \frac{\frac{C_i + 0.5}{L_i}}{\sqrt[n]{\frac{(C_1+0.5)}{L_1} \times \frac{(C_2+0.5)}{L_2} \times \ldots \times \frac{(C_n+0.5)}{L_n}}} \right) \quad (3)$$

### Test of a selective Mcr inhibitor on culture activity

Duplicates of C₆- and C₁₄-oxidizing culture were supplied with substrate and 5 mM (final concentration) BES. A control culture was supplied with substrate but not with BES. Cultures were incubated at 70 °C and sulfide concentrations were measured until the control cultures had reached >15 mM sulfide.

### Metabolite extraction

Metabolite samples were collected at sulfide levels of 10–14 mM. An 80 ml culture sample of each substrate was pelleted via centrifugation (15 min, 3,100 × *g*, 4 °C). Supernatants were removed, pellets were resuspended in 1 ml of acetonitrile:methanol:water (2:2:1 v/v/v) and transferred to bead-beating tubes. Samples were agitated for 15 min on a rotor with vortex adapter at maximum speed. Samples were centrifuged for 20 min at 10,000 × *g* at 4 °C. Clear supernatants were stored at 4 °C.

### Synthesis of authentic alkyl-CoM standards

Coenzyme M (sodium 2-mercaptoethanesulfonate) (0.1 g) was dissolved in 2 ml 25% (v:v) ammonium hydroxide solution and twice the molar amount of bromoalkane was added. We acquired 2- and 3-bromohexane from Tokyo Chemical, and 2-bromotetradecane from Alfa Aesar. Vials were incubated for 6 h at r.t. under gentle shaking on a rotor with vortex adapter. The clear upper phase (1 ml) was collected and stored at 4 °C.

### Mass spectrometry of culture extracts and standards

Culture extracts and standards were analysed using high-resolution accurate-mass mass spectrometry on a Bruker maXis plus quadrupole time-of-flight (QTOF) mass spectrometer (Bruker) connected to a Thermo Dionex Ultimate 3000RS UHPLC system (Thermo Fisher) via an electrospray ionization (ESI) ion source. Sample aliquots were evaporated under a nitrogen stream and re-dissolved in a methanol:water (1:1 v/v) mixture before injection. A 10 µl aliquot of the metabolites was injected and separated on an Acclaim C30 reversed phase column (Thermo Fisher; 3.0 × 250 mm, 3 µm particle size) set to 40 °C using a flow rate of 0.3 ml min⁻¹ and the following gradient of eluent A (acetonitrile:water:formic acid, 5:95:0.1 v/v/v) and eluent B (2-propanol:acetonitrile:formic acid, 90:10:0.1 v/v/v): 0% B at 0 min, then ramp to 100% B at 30 min, hold at 100% B until 50 min, followed by re-equilibration at 0% B from 51 min to the end of the analysis at 60 min to prepare the column for the next analysis. The ESI source was set to the following parameters: capillary voltage 4,500 V, end plate offset 500 V, nebulizer pressure 0.8 bar, dry gas flow 4 l min⁻¹, dry gas heater 200 °C. The QTOF was set to acquire full scan spectra in a mass range of *m/z* 50–600 in negative ionization mode. The C₁₄ culture extract was additionally analysed in tandem mass spectrometry mode, and mass spectra of the fragmentation products of *m/z* 337.1877 isolated in a window of 3 Da and fragmented with 35 eV were acquired. Every analysis was mass-calibrated to reach mass accuracy of 1–3 ppm by loop injection of a calibration solution containing sodium formate cluster ions at the end of the analysis during the equilibration phase and using the high-precision calibration algorithm. Data were processed using the Compass DataAnalysis software package v.5.0 (Bruker).

**Table 1 | Overview over substrate range test with *Ca.* Alkanophaga cultures**

| Organism | Originally consumed $C_x$-*n*-alkanes | Culture used as inoculum | Tested $C_x$-*n*-alkanes |
|---|---|---|---|
| *Ca.* Alkanophaga volatiphilum | $C_5$, $C_6$, $C_7$ | $C_6$ | $C_3$, $C_4$, $C_8$, $C_9$, $C_{10}$, $C_{11}$, $C_{12}$, $C_{13}$, $C_{14}$, $C_{15}$, $C_{16}$, $C_{18}$, $C_{20}$ |
| *Ca.* Alkanophaga liquidiphilum | $C_8$, $C_9$, $C_{10}$, $C_{12}$, $C_{14}$ | $C_{14}$ | $C_3$, $C_4$, $C_{11}$, $C_{13}$, $C_{15}$, $C_{16}$, $C_{18}$, $C_{20}$ |

## Substrate range tests

Cultures originally grown with $C_6$ and $C_{14}$ were diluted 1:10 in fresh SRM. Dilutions were supplemented with alkanes between $C_5$ and $C_{14}$ for which growth had not been confirmed yet, and with shorter ($C_3$ and $C_4$) and longer ($C_{16}$-$C_{20}$) alkanes (Table 1).

A negative (inoculated culture without substrate) and a positive (inoculated culture supplied with substrate with which the culture was originally grown) control were also set up. Cultures were incubated at 70 °C and activity was tracked via sulfide measurements. Once sulfide concentrations reached >10 mM, cultures were diluted 1:3 with SRM. The procedure was repeated and incubations that showed sustained activity over two dilutions were considered successful.

## Hydrogen production measurements

$C_6$ and $C_{14}$ cultures were divided into two 20 ml aliquots in 156 ml serum bottles. One aliquot was left untreated, the other one was treated with 10 mM (final concentration) sodium molybdate. Hydrogen was measured by injecting 1 ml of headspace sample into a Peak Performer 1 gas chromatograph (Peak Laboratories). Measurements were taken in 1 h intervals up to 8 h after the start of the experiment. A final measurement round was conducted from 24 h to 30 h in 2 h intervals.

## Test of the effect of addition of hydrogen and formate on culture activity

Two replicates of $C_6$- and $C_{14}$-oxidizing cultures were supplied with substrate and with 10% $H_2$ in the headspace or 10 mM (final concentration) sodium formate in the medium. A control culture was supplied only with substrate. Cultures were incubated at 70 °C and sulfide concentrations were measured until the control cultures had reached ≥15 mM sulfide.

## Transmission electron microscopy

$C_6$ and $C_{14}$ culture (100 ml) were collected at 1,000 × *g* using a Stat Spin Microprep 2 table top centrifuge. Cells were transferred to aluminium platelets (150 µm depth) containing 1-hexadecene[94]. Platelets were frozen using a Leica EM HPM100 high-pressure freezer (Leica). Frozen samples were transferred to a Leica EM AFS2 automatic freeze substitution unit and substituted at −90 °C in a solution containing anhydrous acetone and 0.1% tannic acid for 24 h, and in anhydrous acetone, 2% $OsO_4$ and 0.5% anhydrous glutaraldehyde (Electron Microscopical Science) for a further 8 h. After further incubation over 20 h at −20 °C, samples were warmed to +4 °C and subsequently washed with anhydrous acetone. Samples were embedded at room temperature in Agar 100 (Epon 812 equivalent) at 60 °C for 24 h. Thin sections (80 nm) were counterstained using Reynolds lead citrate solution for 7 s and examined using a Talos L120C microscope (Thermo Fisher).

## Temperature range tests

Aliquots of $C_6$- and $C_{14}$-oxidizing cultures were diluted 1:6, supplied with substrate and incubated at 60–90 °C in 5 °C increments. Sulfide production was tracked until the 70 °C cultures had reached >10 mM sulfide.

## Availability of biological materials

Official culture collections do not accept syntrophic enrichment cultures, but G.W. will maintain the cultures. Non-profit organizations can obtain samples upon request.

## Reporting summary

Further information on research design is available in the Nature Portfolio Reporting Summary linked to this article.

## Data availability

The following databases were used in this study: SILVA SSU reference database (v.138.1; https://www.arb-silva.de/documentation/release-1381/), NCBI COGs (https://www.ncbi.nlm.nih.gov/research/cog-project/), KEGG (https://www.genome.jp/kegg/kegg1.html), Pfam (https://www.ebi.ac.uk/interpro/), KOfam (https://www.genome.jp/tools/kofamkoala/) plus alignments in ref. 33 (*mer*: Fig04B; *metF*: Fig05C of Supplement S1; https://doi.org/10.1371/journal.pbio.3001508.s017). MAGs of *Ca.* Alkanophaga (*Ca.* A. volatiphilum: BioSample SAMN29995624, genome accession: JAPHEE000000000; *Ca.* A. liquidiphilum: SAMN29995625, JAPHEF000000000) and *Ca.* Thermodesulfobacterium syntrophicum (SAMN29995626, JAPHEG000000000), the raw reads from short-read metagenome and transcriptome sequencing, the coassembly of the $C_6$-$C_{14}$ samples, and the single assemblies of the original slurry and the $C_5$ samples (SAMN30593190, Sequence Read Archive (SRA) accessions SRR22214785-SRR22214804) are accessible under BioProject PRJNA862876. The mass spectrometry runs for the detection of alkyl-CoMs have been deposited to the EMBL-EBI MetaboLights database[95] with the identifier MTBLS7727. Source data are provided with this paper.

## Code availability

The workflow for metagenome and transcriptome analysis, and the scripts for targeted reassembly and for the search of CxxCH motifs are available under https://github.com/zehanna/MCA70_analysis. Further inquiries about bioinformatic analyses may be directed to the corresponding authors.

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

## Acknowledgements

This study was funded by the DFG under Germany's Excellence Initiative/Strategy through the Clusters of Excellence EXC 2077 'The Ocean Floor—Earth's Uncharted Interface' (project no. 390741601), the Andreas Rühl Foundation and the Max Planck Society. R.L.-P. was funded by a Juan de la Cierva grant (FJC2019-041362-I) from the Spanish Ministerio de Ciencia e Innovación. The Guaymas Basin expedition was supported by the National Science Foundation, Biological Oceanography grant no. 1357238 to A.T. (Collaborative Research: Microbial Carbon cycling and its interactions with Sulfur and Nitrogen transformations in Guaymas Basin hydrothermal sediments). We thank the captain and crew of RV *Atlantis* for their excellent work during the expedition AT42-05; H. Taubner and M. Alisch for analytical measurements; S. Menger for technical support in the laboratory; K. Knittel and A. Ellrott for sharing their experience with CARD-FISH and microscopy; and A. Boetius for fruitful scientific discussions.

## Author contributions

H.Z. and G.W. designed the study. G.W. conducted sampling on board. A.T. planned and organized the cruise. R.L.-P. and D.B.M. supported bioinformatic analyses. D.B.M. and H.Z. designed the specific CARD-FISH probe. J.L. conducted mass spectrometry measurements and analyses. D.R. carried out transmission electron microscopy. H.Z. did cultivation and laboratory experiments as well as omics analyses, and wrote the manuscript with contributions from all co-authors. All authors contributed to the article, agreed to all manuscript contents and to the author list and its order, and approved the submitted version.

## Funding

## Competing interests

The authors declare no competing interests.

## Additional information

**Extended data** is available for this paper at https://doi.org/10.1038/s41564-023-01400-3.

**Correspondence and requests for materials** should be addressed to Hanna Zehnle or Gunter Wegener.

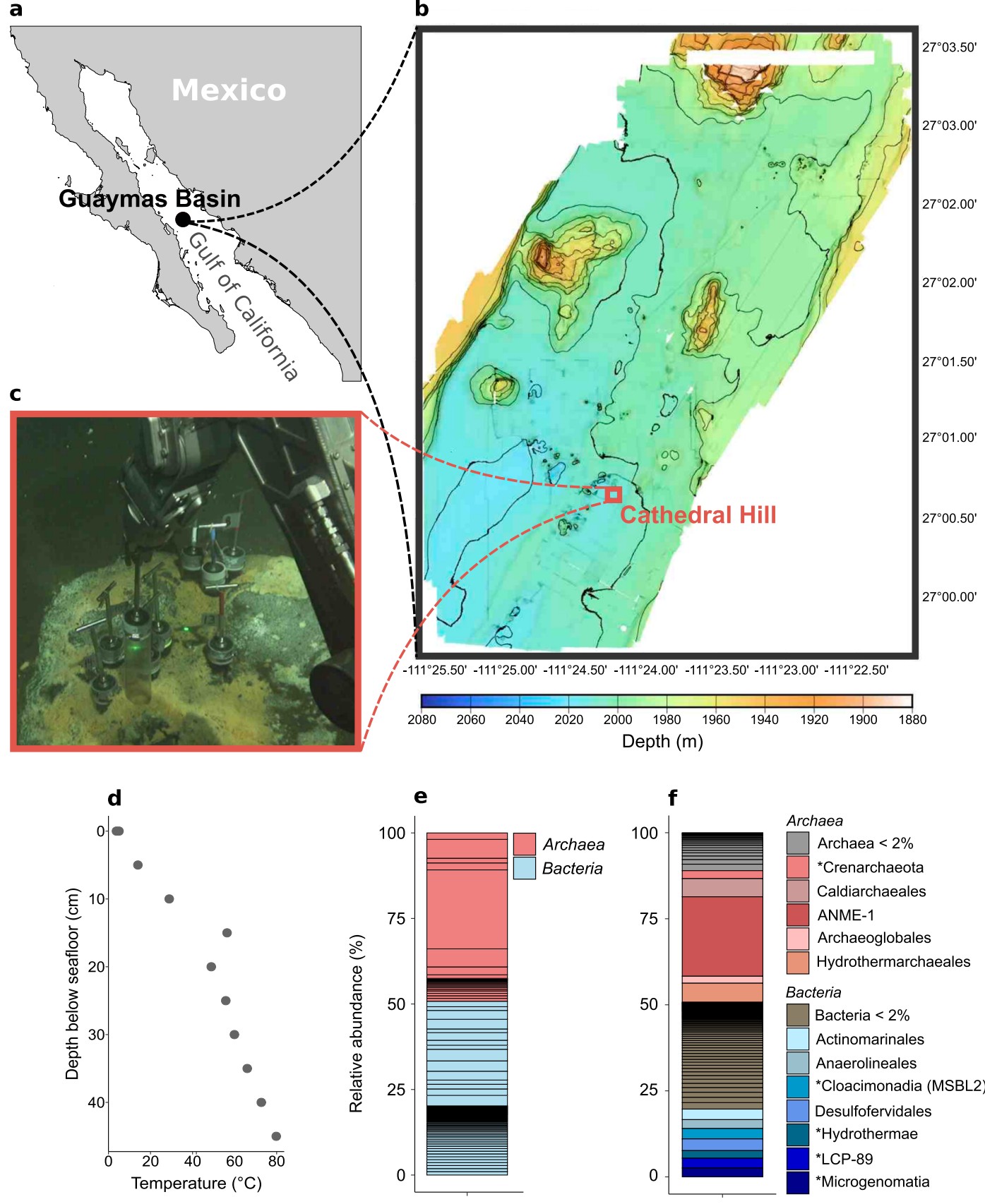

**Extended Data Fig. 1 | See next page for caption.**

**Extended Data Fig. 1 | Sampling site in the Guaymas Basin and microbial community in the original sediment. a**, Location of the Guaymas Basin in the Gulf of California. **b**, Bathymetry of the southern end of the Southern Trough of the Guaymas Basin with the location of the Cathedral Hill hydrothermal vent area. **c**, Sampling of the push core (4991-15) that was used for anoxic cultivations in an area densely covered by orange sulfur-oxidizing *Beggiatoa* mats. **d**, Depth-temperature profile in the sampling site. The temperature was measured using *Alvin*'s heatflow probe. Push cores reached about 30 cm into the sediment, where the temperature approached about 60 °C (sampling site photograph and temperature data courtesy of the Woods Hole Oceanographic Institution, from RV *Atlantis* cruise AT42-05). **e, f** Microbial community in the anoxic sediment slurry prepared from core 4991-15 before starting anoxic incubations based on 16S rRNA gene fragments recruited from the metagenome. **e**, On the domain level, archaea and bacteria each make up around 50%. **f**, Taxonomic groups on order level. For groups with unknown order assignment marked with *, the next known higher taxonomic levels are indicated. An ANME-1 group is abundant within the archaeal fraction while the bacterial fraction is very diverse.

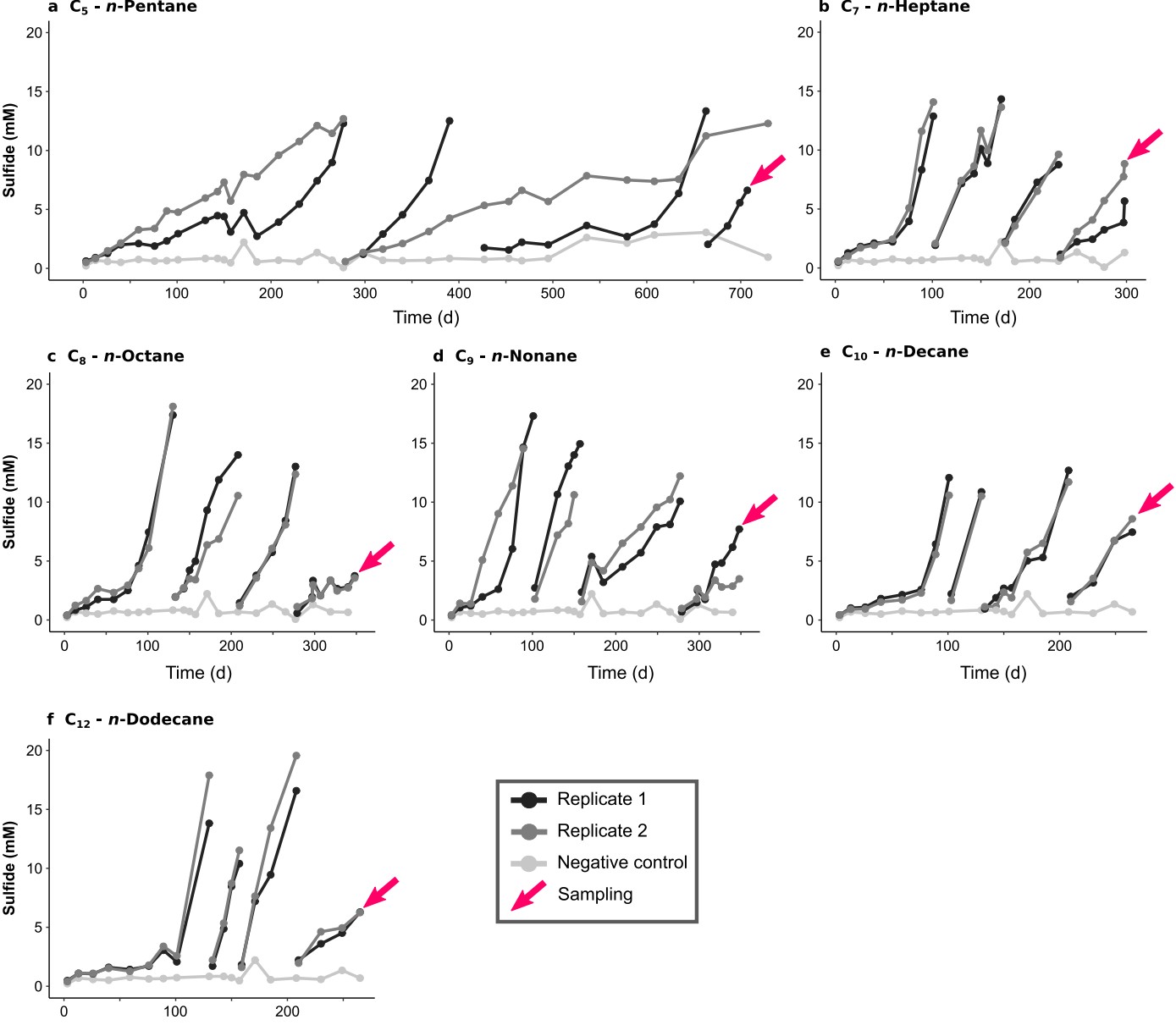

**Extended Data Fig. 2 | Sulfide production in anoxic C₅-C₁₂ *n*-alkane-degrading cultures at 70 °C up to the third dilution.** Each culture was set up as a duplicate. Gaps in sulfide level indicate dilution steps. Pink arrows indicate the sampling points for metagenome and -transcriptome sequencing. Samples were collected after the third dilution from cultures degrading (**a**) *n*-pentane, (**b**) *n*-heptane, (**c**) *n*-octane, (**d**) *n*-nonane, (**e**) *n*-decane, and (**f**) *n*-dodecane. The negative control (light gray line) consisted of a sediment slurry without added substrate.

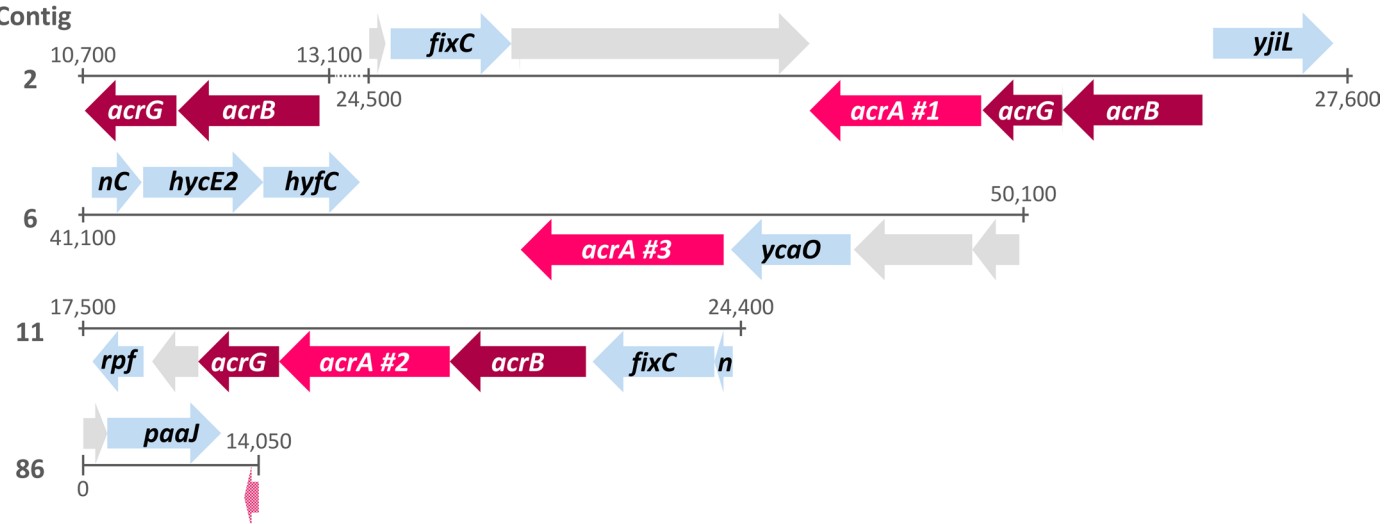

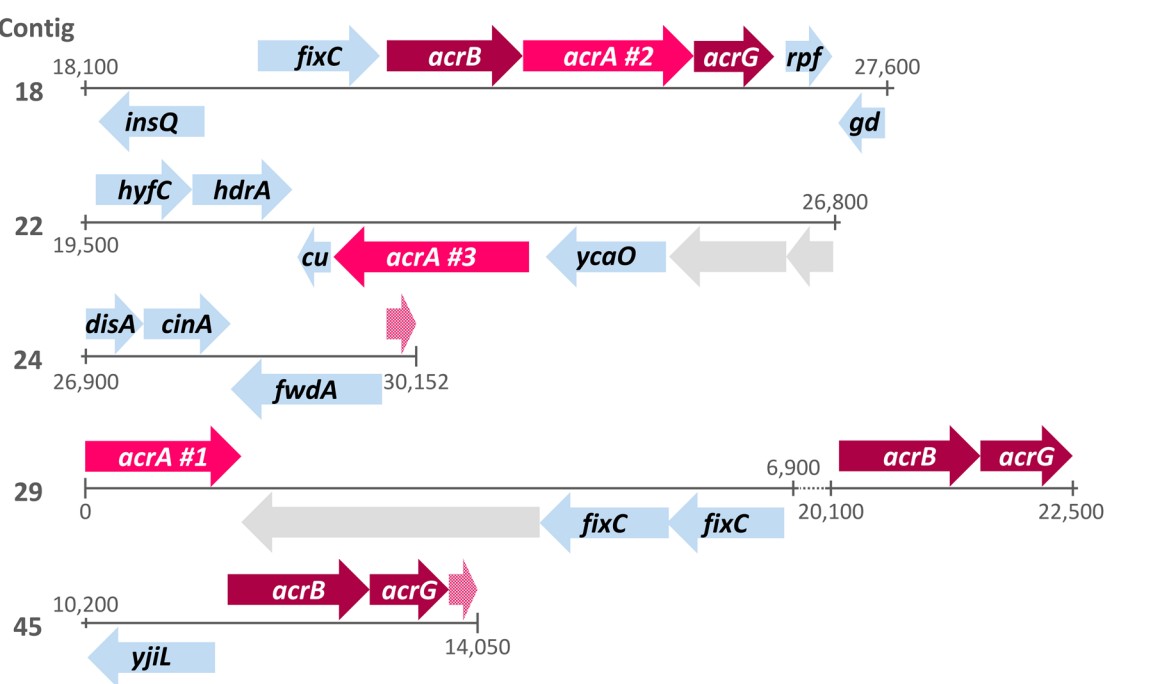

**Extended Data Fig. 3 | Organization of *acr* genes in *Candidatus* Alkanophaga MAGs.** Partial *acrA* genes are shown in light pink, unannotated genes in light gray. Some gene names were shortened to fit the arrows. Genes code for: *acrA*: alkyl-coenzyme M reductase, alpha subunit; *acrB*: alkyl-coenzyme M reductase, beta subunit; *acrG*: alkyl-coenzyme M reductase, gamma subunit; *fixC*: flavoprotein dehydrogenase; *yjiL*: activator of 2-hydroxyglutaryl-CoA dehydratase; *nC*: *nuoC*-NADH:ubiquinone oxidoreductase; *hycE2*: [NiFe]-hydrogenase III large subunit;

*hyfC*: formate hydrogenlyase; *ycaO*: ribosomal protein S12 methylthiotransferase accessory factor; *rpf*: *rpf1*-rRNA maturation protein; *n*: *nuoI*-formate hydrogenlyase subunit 6; *paaJ*: acetyl-CoA acetyltransferase; *insQ*: transposase; *gd*: *gdb1*-glycogen debranching enzyme; *hdrA*: heterodisulfide reductase, subunit A; *cu*: *cutA1*-divalent cation tolerance protein; *disA*: c-di-AMP synthetase; *cinA*: ADP-ribose pyrophosphatase domain of DNA damage- and competence-inducible protein CinA; *fwdA*: formylmethanofuran dehydrogenase subunit A.

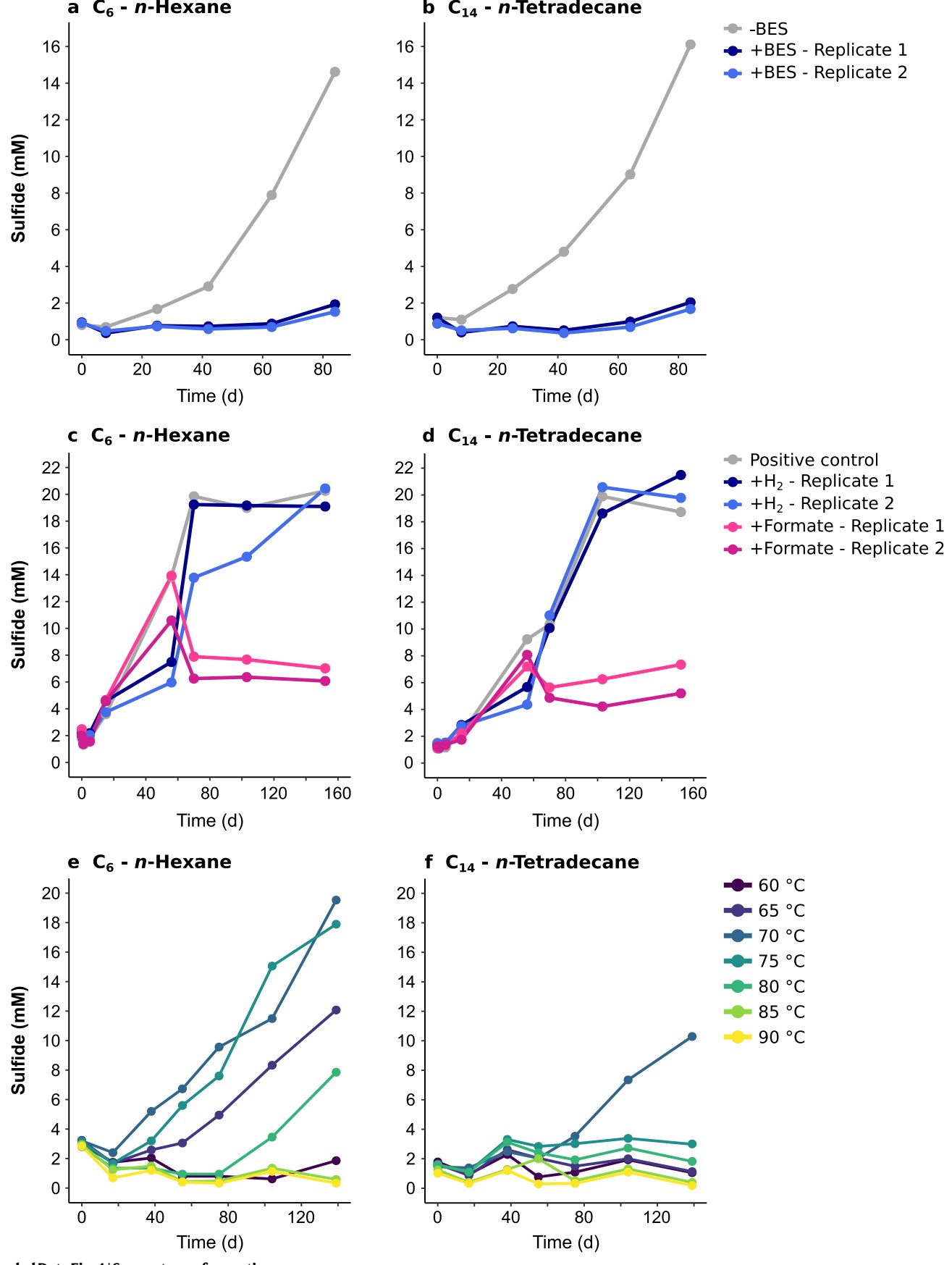

**Extended Data Fig. 4 | See next page for caption.**

**Extended Data Fig. 4 | Sulfide production in *n*-hexane (C<sub>6</sub>)- and *n*-tetradecane (C<sub>14</sub>)-degrading cultures under different conditions. a, b**, Treatment with 2-bromoethanosulfonate (BES). BES (5 mM final concentration) was added to duplicates of $C_6$ (**a**) and $C_{14}$ (**b**) degrading cultures ( + BES). A control culture (-BES) did not receive BES. The inhibition of alkane oxidation by BES corroborates an Acr-based substrate activation. **c,d**, Addition of hydrogen or formate to $C_6$ (**c**) and $C_{14}$ (**d**)-degrading cultures. All cultures were supplied with the original substrate. The addition of 10% $H_2$ into the headspace or 10 mM sodium formate into the medium did not accelerate sulfide production compared to positive controls. **e, f**, Incubation at temperatures between 60 °C and 90 °C. The $C_6$-degrading culture (**e**) grows optimally at 70 °C and 75 °C, while it still shows some activity at slightly lower (65 °C) and slightly higher (80 °C) temperatures. The activity of the $C_{14}$-degrading culture (**f**) seems to be limited to 70 °C.

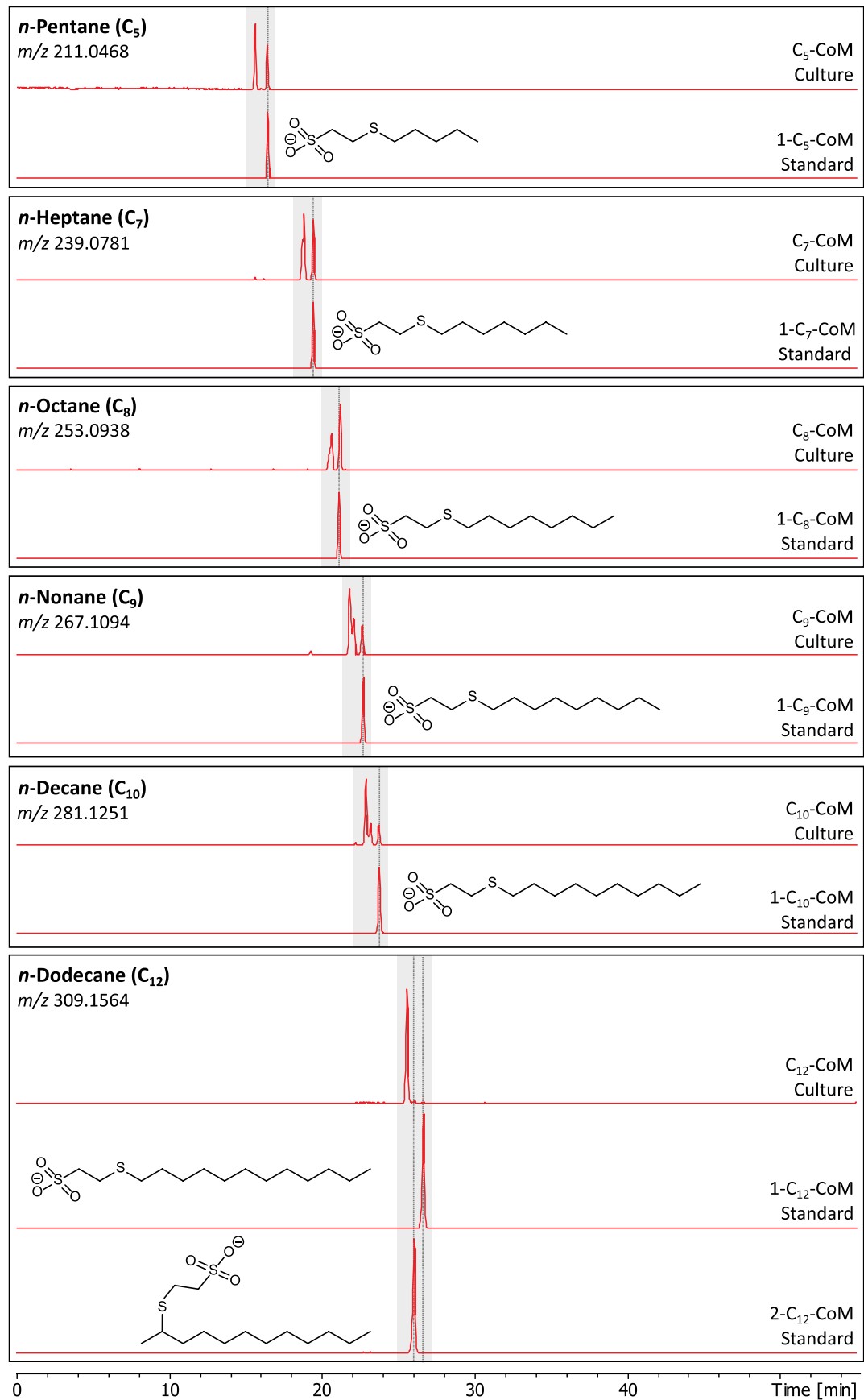

**Extended Data Fig. 5 | See next page for caption.**

**Extended Data Fig. 5 | Detection of alkyl-CoMs in C$_x$-*n*-alkane-degrading cultures.** Samples were separated by liquid chromatography and extracted ion chromatograms (EICs) based on the exact mass of deprotonated ions of the C$_x$-alkyl-CoMs with a window of ±10 mDa were created. Panels show the EICs of culture extracts together with synthetic standards. Dashed vertical lines were added at the retention times of peak maxima of the standards for easier identification of peaks in the culture extracts. Peaks with mass-to-charge ratios (*m/z*) of the respective alkyl-CoM were detected in all cultures. All culture extracts show several peaks, indicating an activation at different carbon atoms. While shorter alkanes are activated to a similar degree at subterminal and terminal positions, longer alkanes are predominantly activated at non-terminal carbon atoms.

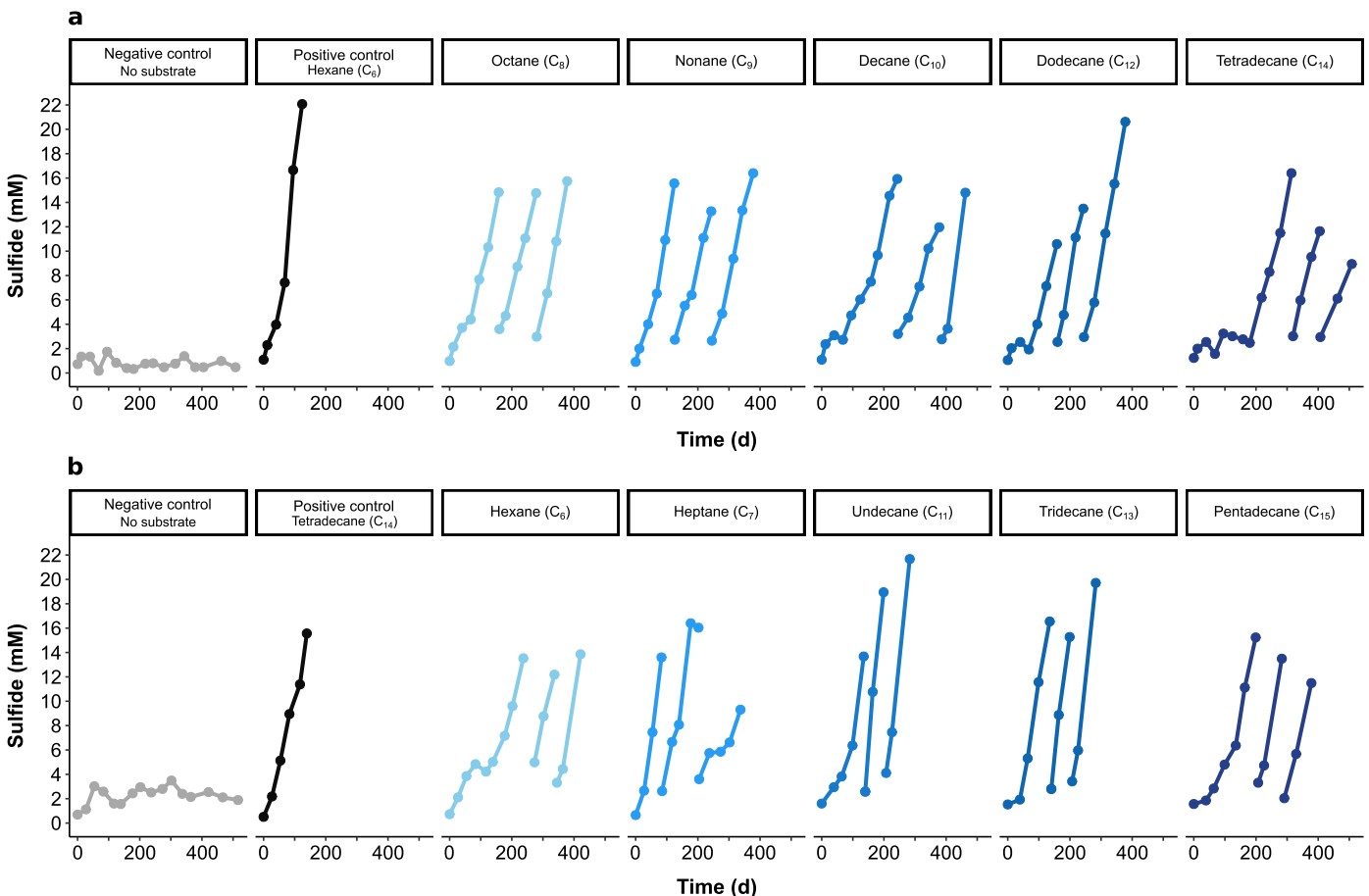

**Extended Data Fig. 6 | Substrate spectra of originally (a) *n*-hexane- and (b) *n*-tetradecane-oxidizing enrichment cultures.** Cultures were diluted into fresh sulfate-reducer medium and supplemented with other *n*-alkanes between $C_3$ and $C_{20}$. Only active cultures are shown. No activity was observed for cultures supplied with $C_3$, $C_4$, $C_{16}$, $C_{18}$, or $C_{20}$.

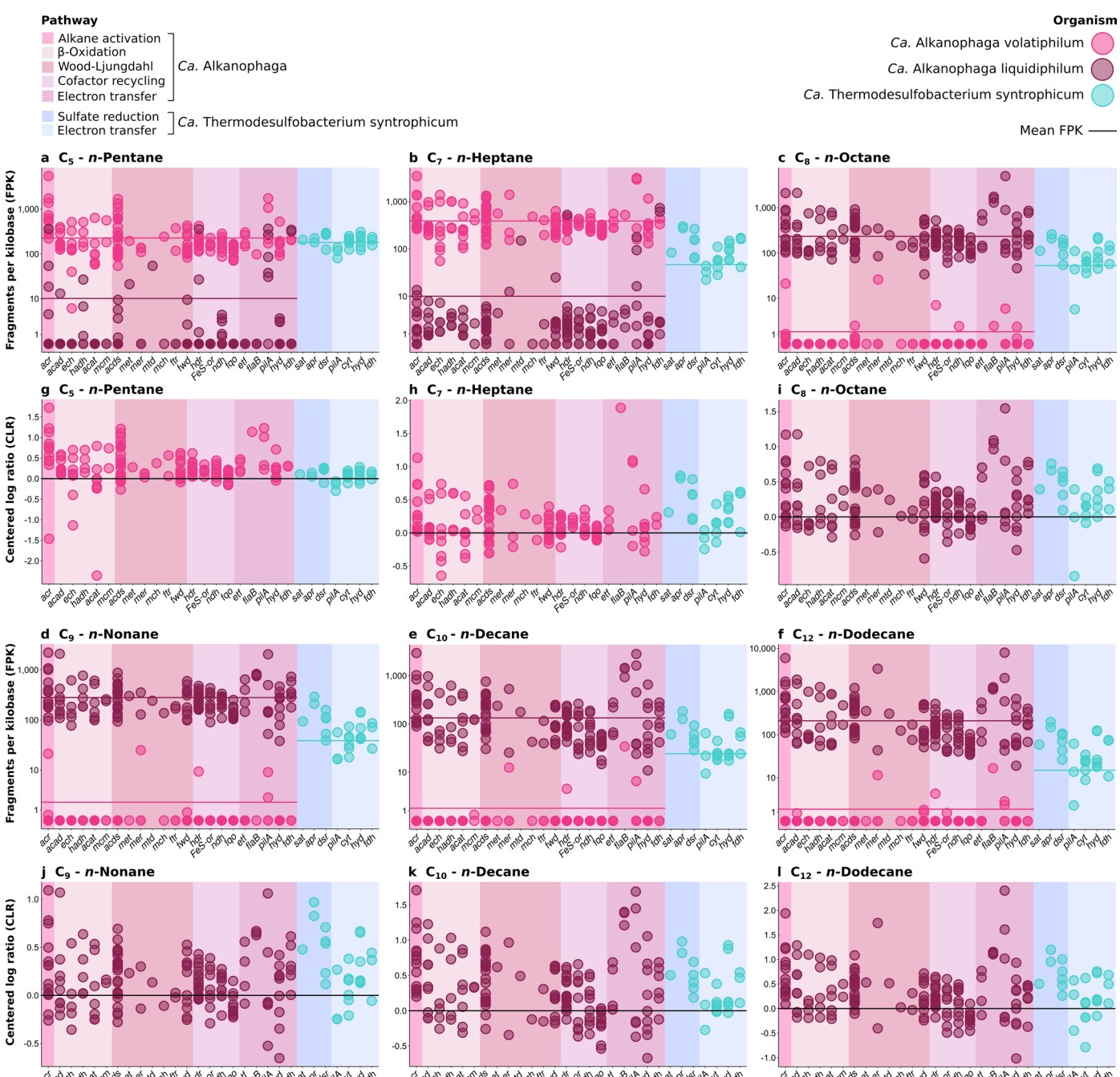

**Extended Data Fig. 7 | Expression of alkane oxidation, sulfate reduction, and related genes in C$_5$-C$_{12}$ $n$-alkane-oxidizing cultures.** Transcriptome reads were mapped to the MAGs of the two *Candidatus* Alkanophaga species and to *Ca.* Thermodesulfobacterium syntrophicum. **a-f**, Fragment counts normalized to gene length (FPK) using a logarithmic y axis. The average gene expression of each organism is indicated as arithmetic mean (sum of all FPK values divided by number of genes) depicted as a horizontal line. **g-l**, Fragment counts normalized as CLR. For simplicity, only values of the more active *Ca.* Alkanophaga species are shown. The x-axis shows the genes encoding: *acr*: alkyl-CoM reductase, *acad*: acyl-CoA dehydrogenase, *ech*: enoyl-CoA hydratase, *hadh*: hydroxyacyl-CoA dehydrogenase, *acat*: acetyl-CoA acetyltransferase, *mcm*: methylmalonyl-

CoA mutase, *acds*: acetyl-CoA decarbonylase/synthase, *met*: 5,10-methylene tetrahydrofolate reductase, *mer*: 5,10-methylene tetrahydromethanopterin (H$_4$MPT) reductase, *mtd*: methylene-H$_4$MPT dehydrogenase, *mch*: methenyl-H$_4$MPT cyclohydrolase, *ftr*: formylmethanofuran-H$_4$MPT formyltransferase, *fwd*: tungsten-containing formylmethanofuran dehydrogenase, *hdr*: heterodisulfide reductase, *FeS-or*: [FeS]-oxidoreductase, *ndh*: NADH dehydrogenase, *fqo*: F$_{420}$H$_2$:quinone oxidoreductase, *etf*: electron transfer flavoprotein, *flaB*: flagellin B, *pilA*: type IV pilin, *hyd*: [NiFe]-hydrogenase, *fdh*: formate dehydrogenase, *sat*: ATP-sulfurylase, *apr*: APS-reductase, *dsr*: dissimilatory sulfite reductase, *cyt*: multi-heme $c$-type cytochrome.

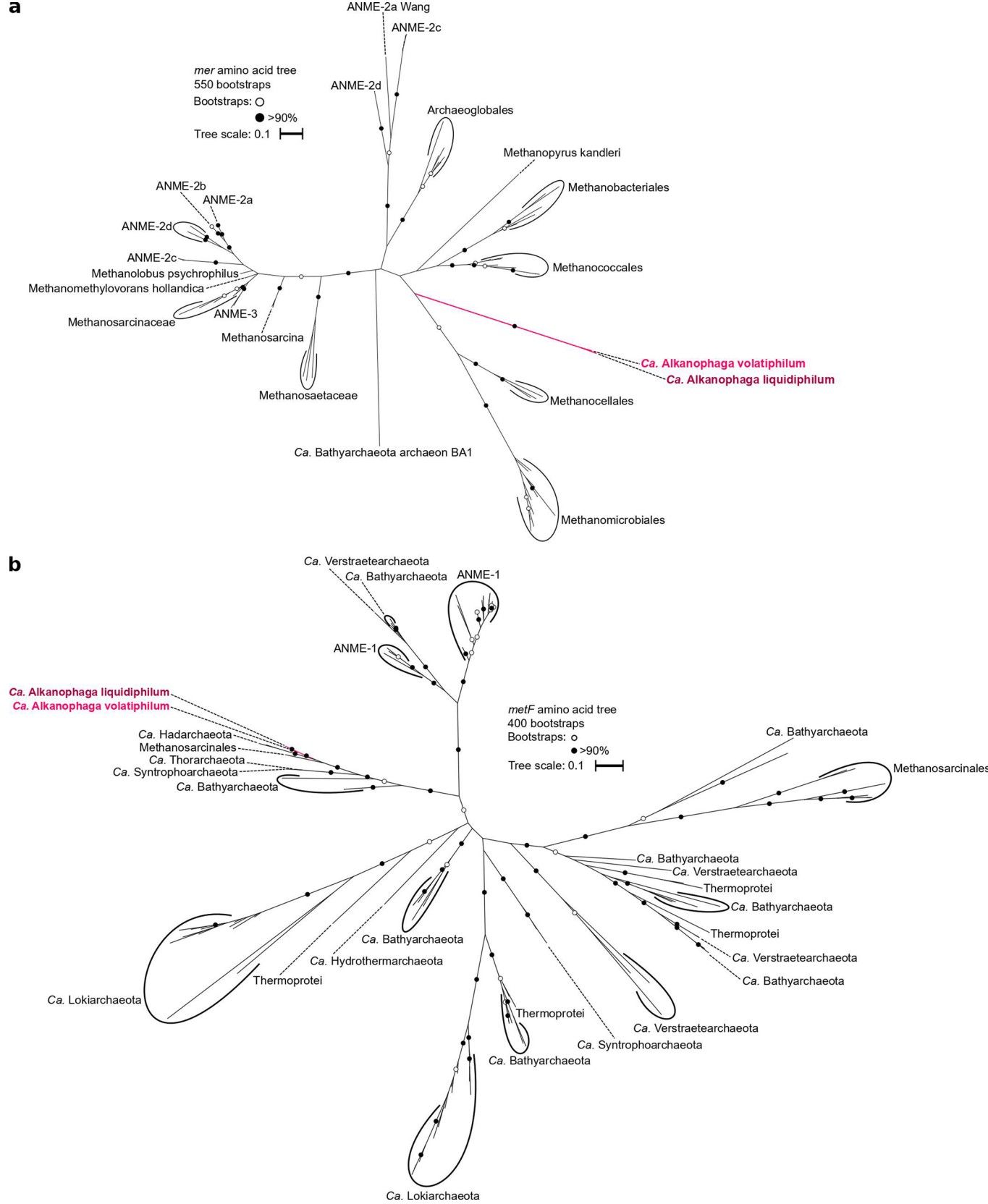

**Extended Data Fig. 8 | Phylogenetic placement of (a) 5,10-methylene-H₄MPT reductase (*mer*) and (b) methylenetetrahydrofolate reductase (*metF*) sequences recovered from *Ca*. Alkanophaga MAGs.** Both *mer* and *metF* sequences of the two *Ca*. Alkanophaga species are highly similar to each other. The *mer* sequences, which distinguish *Ca*. Alkanophaga in the class Syntrophoarchaeia, might originate from the ancestor of Methanocellales, while *metF* sequences cluster near those of close relatives *Ca*. Syntrophoarchaeales.

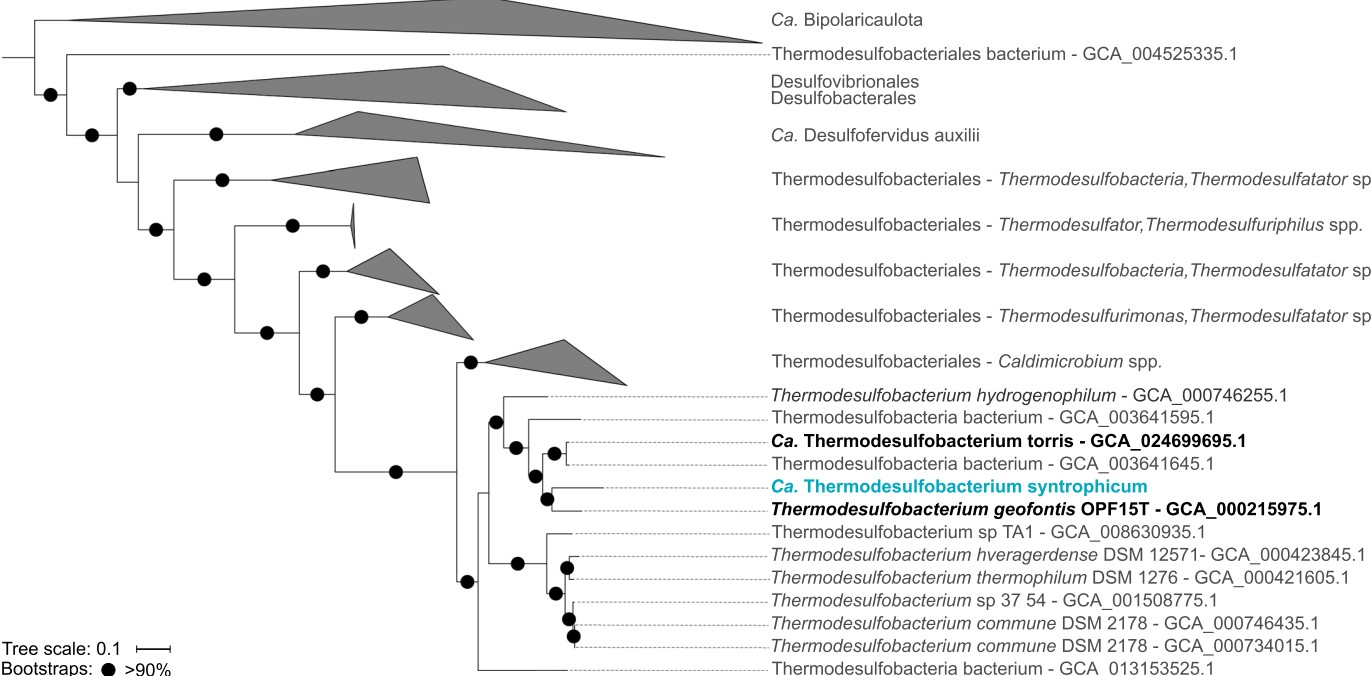

**Extended Data Fig. 9 | Phylogenomic placement of *Candidatus* Thermodesulfobacterium syntrophicum based on the concatenated alignment of 71 bacterial single copy core genes.** *Ca.* T. syntrophicum is closely related to the already cultured *Thermodesulfobacterium geofontis* (OPF15T) and to *Ca.* Thermodesulfobacterium torris, which functions as partner bacterium in the thermophilic anaerobic oxidation of methane. The outgroup consists of members of the candidate phylum Bipolauricaulota. The tree scale bar indicates 10% sequence divergence.

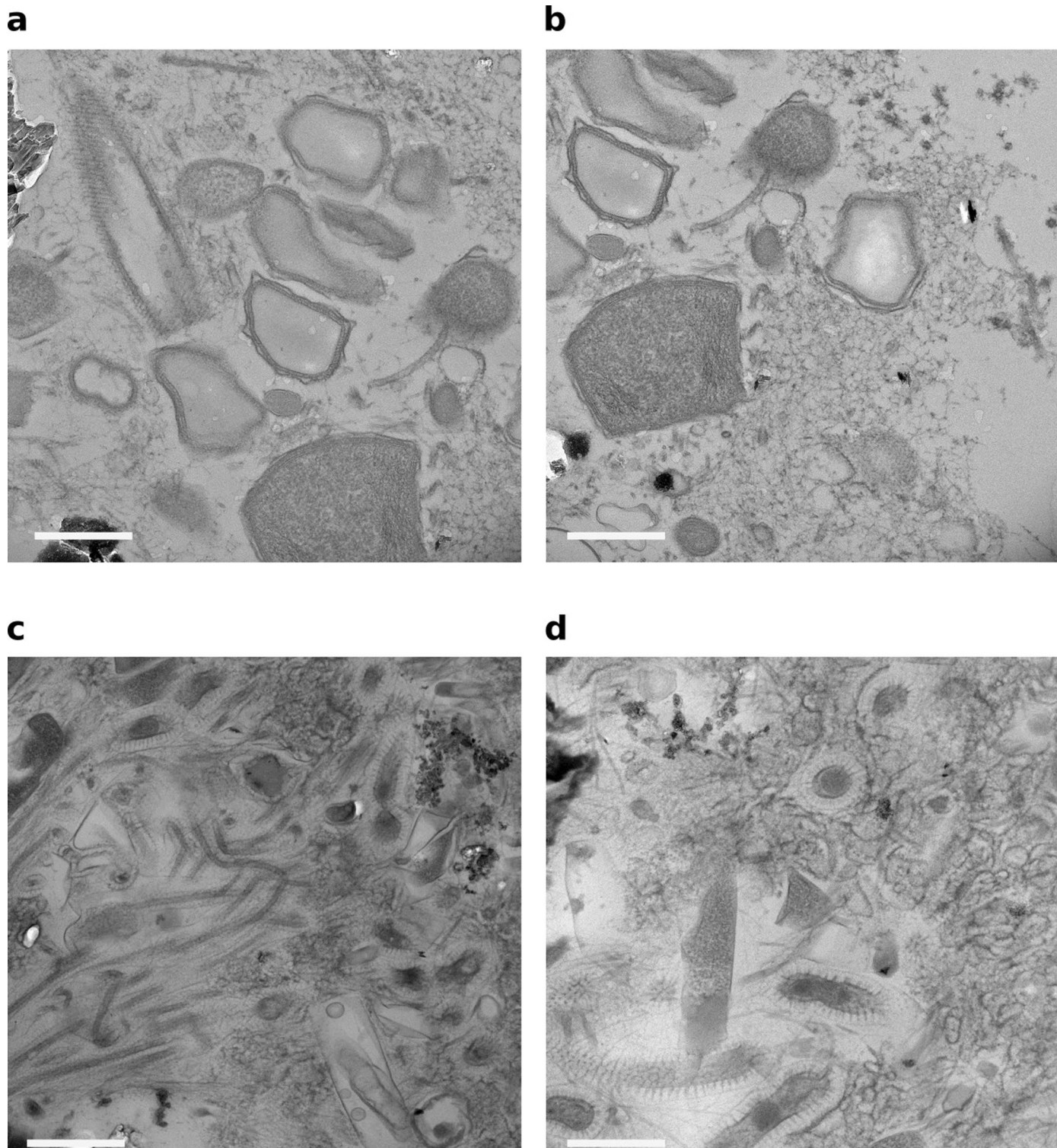

**a**

**b**

**c**

**d**

**Extended Data Fig. 10 | Transmission electron micrographs of EPON 812-embedded thin-sections of (a,b) C$_6$- and (c,d) C$_{14}$-$n$-alkane-degrading culture samples.** The scale bar indicates 0.5 µm. The experiment was run once with one biological replicate per sample. Images are representative for > 5 recorded images per sample.

Gunter Wegener

# Reporting Summary

## Statistics

For all statistical analyses, confirm that the following items are present in the figure legend, table legend, main text, or Methods section.

| n/a | Confirmed | |
|---|---|---|
| ☐ | ☒ | The exact sample size (*n*) for each experimental group/condition, given as a discrete number and unit of measurement |
| ☐ | ☒ | A statement on whether measurements were taken from distinct samples or whether the same sample was measured repeatedly |
| ☒ | ☐ | The statistical test(s) used AND whether they are one- or two-sided<br>*Only common tests should be described solely by name; describe more complex techniques in the Methods section.* |
| ☒ | ☐ | A description of all covariates tested |
| ☒ | ☐ | A description of any assumptions or corrections, such as tests of normality and adjustment for multiple comparisons |
| ☐ | ☒ | A full description of the statistical parameters including central tendency (e.g. means) or other basic estimates (e.g. regression coefficient) AND variation (e.g. standard deviation) or associated estimates of uncertainty (e.g. confidence intervals) |
| ☒ | ☐ | For null hypothesis testing, the test statistic (e.g. *F*, *t*, *r*) with confidence intervals, effect sizes, degrees of freedom and *P* value noted<br>*Give P values as exact values whenever suitable.* |
| ☒ | ☐ | For Bayesian analysis, information on the choice of priors and Markov chain Monte Carlo settings |
| ☒ | ☐ | For hierarchical and complex designs, identification of the appropriate level for tests and full reporting of outcomes |
| ☒ | ☐ | Estimates of effect sizes (e.g. Cohen's *d*, Pearson's *r*), indicating how they were calculated |

*Our web collection on statistics for biologists contains articles on many of the points above.*

## Software and code

Policy information about availability of computer code

| | |
|---|---|
| Data collection | CARD-FISH images: AxioVision software (v4.8; Axiophot II imaging with AxioCamMR camera; Zeiss, Oberkochen, Germany); Mass spectrometry: Compass DataAnalysis software (v5.0; Bruker Daltonics, Bremen, Germany) |
| Data analysis | BBMap v38.79; BBMap v38.87; SPAdes v3.14.0; anvi'o v7; Bowtie2 v2.3.2; SAMtools v1.5; Centrifuge v1.0.2-beta; CheckM v1.1.3; GTDB-Tk v1.5.1; Prokka v1.14.6; CoverM v0.6.1; Minimap2 v2.21; FastANI v1.32; CompareM v0.1.2; ARB v7.1; featureCounts v1.4.6-p5; MUSCLE v5.1; RAxML v8.2.12; RAxML v8.2.4; MAFFT v7.475; SeaView v5; phyloFlash v3.4.1; ImageJ v1.49 |

For manuscripts utilizing custom algorithms or software that are central to the research but not yet described in published literature, software must be made available to editors and reviewers. We strongly encourage code deposition in a community repository (e.g. GitHub). See the Nature Portfolio guidelines for submitting code & software for further information.

## Data

Policy information about availability of data

All manuscripts must include a data availability statement. This statement should provide the following information, where applicable:
- Accession codes, unique identifiers, or web links for publicly available datasets
- A description of any restrictions on data availability
- For clinical datasets or third party data, please ensure that the statement adheres to our policy

The following databases were used in this study: SILVA SSU reference database (version 138.1; https://www.arb-silva.de/documentation/release-1381/), NCBI COGs

(https://www.ncbi.nlm.nih.gov/research/cog-project/), KEGG (https://www.genome.jp/kegg/kegg1.html), Pfam (https://www.ebi.ac.uk/interpro/), KOfam (https://www.genome.jp/tools/kofamkoala/) plus the publicly available alignments by Chadwick et al (mer: Fig04B; metF: Fig05C of Supplement S1; https://doi.org/10.1371/journal.pbio.3001508.s017).

MAGs of Ca. Alkanophaga (Ca. A. volatiphilum: BioSample SAMN29995624, genome accession: JAPHEE000000000; Ca. A. liquidiphilum: SAMN29995625, JAPHEF000000000) and Ca. Thermodesulfobacterium syntrophicum (SAMN29995626, JAPHEG000000000), the raw reads from short-read metagenome and -transcriptome sequencing, the coassembly of the C6 to C14 samples, and the single assemblies of the original slurry and the C5 samples (SAMN30593190, Sequence Read Archive (SRA) accessions SRR22214785-SRR22214804) have been deposited under BioProject PRJNA862876.

## Human research participants

Policy information about studies involving human research participants and Sex and Gender in Research.

| | |
|---|---|
| Reporting on sex and gender | n.a. |
| Population characteristics | n.a. |
| Recruitment | n.a. |
| Ethics oversight | n.a. |

Note that full information on the approval of the study protocol must also be provided in the manuscript.

# Field-specific reporting

Please select the one below that is the best fit for your research. If you are not sure, read the appropriate sections before making your selection.

☐ Life sciences   ☐ Behavioural & social sciences   ☒ Ecological, evolutionary & environmental sciences

For a reference copy of the document with all sections, see nature.com/documents/nr-reporting-summary-flat.pdf

# Life sciences study design

All studies must disclose on these points even when the disclosure is negative.

| | |
|---|---|
| Sample size | Describe how sample size was determined, detailing any statistical methods used to predetermine sample size OR if no sample-size calculation was performed, describe how sample sizes were chosen and provide a rationale for why these sample sizes are sufficient. |
| Data exclusions | Describe any data exclusions. If no data were excluded from the analyses, state so OR if data were excluded, describe the exclusions and the rationale behind them, indicating whether exclusion criteria were pre-established. |
| Replication | Describe the measures taken to verify the reproducibility of the experimental findings. If all attempts at replication were successful, confirm this OR if there are any findings that were not replicated or cannot be reproduced, note this and describe why. |
| Randomization | Describe how samples/organisms/participants were allocated into experimental groups. If allocation was not random, describe how covariates were controlled OR if this is not relevant to your study, explain why. |
| Blinding | Describe whether the investigators were blinded to group allocation during data collection and/or analysis. If blinding was not possible, describe why OR explain why blinding was not relevant to your study. |

# Behavioural & social sciences study design

All studies must disclose on these points even when the disclosure is negative.

| | |
|---|---|
| Study description | Briefly describe the study type including whether data are quantitative, qualitative, or mixed-methods (e.g. qualitative cross-sectional, quantitative experimental, mixed-methods case study). |
| Research sample | State the research sample (e.g. Harvard university undergraduates, villagers in rural India) and provide relevant demographic information (e.g. age, sex) and indicate whether the sample is representative. Provide a rationale for the study sample chosen. For studies involving existing datasets, please describe the dataset and source. |
| Sampling strategy | Describe the sampling procedure (e.g. random, snowball, stratified, convenience). Describe the statistical methods that were used to predetermine sample size OR if no sample-size calculation was performed, describe how sample sizes were chosen and provide a rationale for why these sample sizes are sufficient. For qualitative data, please indicate whether data saturation was considered, and what criteria were used to decide that no further sampling was needed. |

| | |
|---|---|
| Data collection | *Provide details about the data collection procedure, including the instruments or devices used to record the data (e.g. pen and paper, computer, eye tracker, video or audio equipment) whether anyone was present besides the participant(s) and the researcher, and whether the researcher was blind to experimental condition and/or the study hypothesis during data collection.* |
| Timing | *Indicate the start and stop dates of data collection. If there is a gap between collection periods, state the dates for each sample cohort.* |
| Data exclusions | *If no data were excluded from the analyses, state so OR if data were excluded, provide the exact number of exclusions and the rationale behind them, indicating whether exclusion criteria were pre-established.* |
| Non-participation | *State how many participants dropped out/declined participation and the reason(s) given OR provide response rate OR state that no participants dropped out/declined participation.* |
| Randomization | *If participants were not allocated into experimental groups, state so OR describe how participants were allocated to groups, and if allocation was not random, describe how covariates were controlled.* |

# Ecological, evolutionary & environmental sciences study design

All studies must disclose on these points even when the disclosure is negative.

| | |
|---|---|
| Study description | This is an exploratory study targeting oil-degrading microorganisms. We sampled areas / environments / physicochemical gradients that have been used for prior cultivation attempts. These sediments were used for the described cultivation procedure. |
| Research sample | This study based on a single sediment core (Alvin dive 4991, core 15) retrieved from a sediment-hosted hydrothermal vents in the Guaymas Basin. |
| Sampling strategy | This study involved exploratory sampling of seafloor sediments. We sampled an area densely covered by microbial mats. The sulfide oxidation activation pointed towards strong, alkane-dependent sulfide production in the sediment. Temperature measurements revealed the potential for thermophilic microorganisms. |
| Data collection | Field data was collected with the research submarine Alvin, operated from the research vessel RV Atlantis during cruise AT42-05 |
| Timing and spatial scale | The sampling was part of a two weeks sampling effort in the Guaymas Basin in November 2018. This study bases however on a single sample, collected from a mat-covered area at the Cathedral Hill hydrothermal vent complex on November 17, 2018. |
| Data exclusions | No data was excluded. |
| Reproducibility | We performed all cultivation attempts in duplicates. All duplicate pairs produced highly similar results. |
| Randomization | This is an exploratory study targeting novel microbial processes and did not require randomization. |
| Blinding | No |

Did the study involve field work?  ☒ Yes  ☐ No

## Field work, collection and transport

| | |
|---|---|
| Field conditions | This study was performed in stable deep-sea waters (water depth 2013 m). The water temperature was 4°C. |
| Location | Guaymas Basin, Gulf of California, Mexico (27°0.6848N, 111°24.2708W). |
| Access & import/export | The Guaymas Basin was accessed with the Reseach Vessel Atlantis and the research submarine Alvin. Sampling and export of samples was done under the sampling license / Permiso de Pesca de Fomento a Extranjeros No. PRFE/DPOPA-207/18 given to Prof. Andreas Teske. |
| Disturbance | We took only single cores in a larger sampling area. No macrofauna was sampled for this study. The microbial communities in this area will recover rapidly. |

# Reporting for specific materials, systems and methods

We require information from authors about some types of materials, experimental systems and methods used in many studies. Here, indicate whether each material, system or method listed is relevant to your study. If you are not sure if a list item applies to your research, read the appropriate section before selecting a response.

## Materials & experimental systems

| n/a | Involved in the study |
|-----|----------------------|
| ☒ | Antibodies |
| ☒ | Eukaryotic cell lines |
| ☒ | Palaeontology and archaeology |
| ☒ | Animals and other organisms |
| ☒ | Clinical data |
| ☒ | Dual use research of concern |

## Methods

| n/a | Involved in the study |
|-----|----------------------|
| ☒ | ChIP-seq |
| ☒ | Flow cytometry |
| ☒ | MRI-based neuroimaging |

# Antibodies

| Antibodies used | *Describe all antibodies used in the study; as applicable, provide supplier name, catalog number, clone name, and lot number.* |
|-----------------|---------------------------------------------------------------------------------------------------------------------------------|
| Validation | *Describe the validation of each primary antibody for the species and application, noting any validation statements on the manufacturer's website, relevant citations, antibody profiles in online databases, or data provided in the manuscript.* |

# Eukaryotic cell lines

Policy information about cell lines and Sex and Gender in Research

| Cell line source(s) | *State the source of each cell line used and the sex of all primary cell lines and cells derived from human participants or vertebrate models.* |
|---------------------|-----------------------------------------------------------------------------------------------------------------------------------------------|
| Authentication | *Describe the authentication procedures for each cell line used OR declare that none of the cell lines used were authenticated.* |
| Mycoplasma contamination | *Confirm that all cell lines tested negative for mycoplasma contamination OR describe the results of the testing for mycoplasma contamination OR declare that the cell lines were not tested for mycoplasma contamination.* |
| Commonly misidentified lines (See ICLAC register) | *Name any commonly misidentified cell lines used in the study and provide a rationale for their use.* |

# Palaeontology and Archaeology

| Specimen provenance | *Provide provenance information for specimens and describe permits that were obtained for the work (including the name of the issuing authority, the date of issue, and any identifying information). Permits should encompass collection and, where applicable, export.* |
|---------------------|---------------------------------------------------------------------------------------------------------------------------------------------|
| Specimen deposition | *Indicate where the specimens have been deposited to permit free access by other researchers.* |
| Dating methods | *If new dates are provided, describe how they were obtained (e.g. collection, storage, sample pretreatment and measurement), where they were obtained (i.e. lab name), the calibration program and the protocol for quality assurance OR state that no new dates are provided.* |

☐ Tick this box to confirm that the raw and calibrated dates are available in the paper or in Supplementary Information.

| Ethics oversight | *Identify the organization(s) that approved or provided guidance on the study protocol, OR state that no ethical approval or guidance was required and explain why not.* |
|------------------|----------------------------------------------------------------------------------------------------------------------------------------------|

Note that full information on the approval of the study protocol must also be provided in the manuscript.

# Animals and other research organisms

Policy information about studies involving animals; ARRIVE guidelines recommended for reporting animal research, and Sex and Gender in Research

| Laboratory animals | *For laboratory animals, report species, strain and age OR state that the study did not involve laboratory animals.* |
|--------------------|-------------------------------------------------------------------------------------------------------------------|
| Wild animals | *Provide details on animals observed in or captured in the field; report species and age where possible. Describe how animals were caught and transported and what happened to captive animals after the study (if killed, explain why and describe method; if released, say where and when) OR state that the study did not involve wild animals.* |
| Reporting on sex | *Indicate if findings apply to only one sex; describe whether sex was considered in study design, methods used for assigning sex. Provide data disaggregated for sex where this information has been collected in the source data as appropriate; provide overall numbers in this Reporting Summary. Please state if this information has not been collected. Report sex-based analyses where performed, justify reasons for lack of sex-based analysis.* |

| Field-collected samples | *For laboratory work with field-collected samples, describe all relevant parameters such as housing, maintenance, temperature, photoperiod and end-of-experiment protocol OR state that the study did not involve samples collected from the field.* |
| Ethics oversight | *Identify the organization(s) that approved or provided guidance on the study protocol, OR state that no ethical approval or guidance was required and explain why not.* |

Note that full information on the approval of the study protocol must also be provided in the manuscript.

# Clinical data

Policy information about clinical studies

All manuscripts should comply with the ICMJE guidelines for publication of clinical research and a completed CONSORT checklist must be included with all submissions.

| Clinical trial registration | *Provide the trial registration number from ClinicalTrials.gov or an equivalent agency.* |
| Study protocol | *Note where the full trial protocol can be accessed OR if not available, explain why.* |
| Data collection | *Describe the settings and locales of data collection, noting the time periods of recruitment and data collection.* |
| Outcomes | *Describe how you pre-defined primary and secondary outcome measures and how you assessed these measures.* |

# Dual use research of concern

Policy information about dual use research of concern

## Hazards

Could the accidental, deliberate or reckless misuse of agents or technologies generated in the work, or the application of information presented in the manuscript, pose a threat to:

No | Yes
☐ ☐ Public health
☐ ☐ National security
☐ ☐ Crops and/or livestock
☐ ☐ Ecosystems
☐ ☐ Any other significant area

## Experiments of concern

Does the work involve any of these experiments of concern:

No | Yes
☐ ☐ Demonstrate how to render a vaccine ineffective
☐ ☐ Confer resistance to therapeutically useful antibiotics or antiviral agents
☐ ☐ Enhance the virulence of a pathogen or render a nonpathogen virulent
☐ ☐ Increase transmissibility of a pathogen
☐ ☐ Alter the host range of a pathogen
☐ ☐ Enable evasion of diagnostic/detection modalities
☐ ☐ Enable the weaponization of a biological agent or toxin
☐ ☐ Any other potentially harmful combination of experiments and agents

# ChIP-seq

## Data deposition

☐ Confirm that both raw and final processed data have been deposited in a public database such as GEO.

☐ Confirm that you have deposited or provided access to graph files (e.g. BED files) for the called peaks.

| Data access links<br>*May remain private before publication.* | *For "Initial submission" or "Revised version" documents, provide reviewer access links. For your "Final submission" document, provide a link to the deposited data.* |
| Files in database submission | *Provide a list of all files available in the database submission.* |

Genome browser session
(e.g. UCSC)

Provide a link to an anonymized genome browser session for "Initial submission" and "Revised version" documents only, to enable peer review. Write "no longer applicable" for "Final submission" documents.

## Methodology

| | |
|---|---|
| Replicates | Describe the experimental replicates, specifying number, type and replicate agreement. |
| Sequencing depth | Describe the sequencing depth for each experiment, providing the total number of reads, uniquely mapped reads, length of reads and whether they were paired- or single-end. |
| Antibodies | Describe the antibodies used for the ChIP-seq experiments; as applicable, provide supplier name, catalog number, clone name, and lot number. |
| Peak calling parameters | Specify the command line program and parameters used for read mapping and peak calling, including the ChIP, control and index files used. |
| Data quality | Describe the methods used to ensure data quality in full detail, including how many peaks are at FDR 5% and above 5-fold enrichment. |
| Software | Describe the software used to collect and analyze the ChIP-seq data. For custom code that has been deposited into a community repository, provide accession details. |

# Flow Cytometry

## Plots

Confirm that:

☐ The axis labels state the marker and fluorochrome used (e.g. CD4-FITC).

☐ The axis scales are clearly visible. Include numbers along axes only for bottom left plot of group (a 'group' is an analysis of identical markers).

☐ All plots are contour plots with outliers or pseudocolor plots.

☐ A numerical value for number of cells or percentage (with statistics) is provided.

## Methodology

| | |
|---|---|
| Sample preparation | Describe the sample preparation, detailing the biological source of the cells and any tissue processing steps used. |
| Instrument | Identify the instrument used for data collection, specifying make and model number. |
| Software | Describe the software used to collect and analyze the flow cytometry data. For custom code that has been deposited into a community repository, provide accession details. |
| Cell population abundance | Describe the abundance of the relevant cell populations within post-sort fractions, providing details on the purity of the samples and how it was determined. |
| Gating strategy | Describe the gating strategy used for all relevant experiments, specifying the preliminary FSC/SSC gates of the starting cell population, indicating where boundaries between "positive" and "negative" staining cell populations are defined. |

☐ Tick this box to confirm that a figure exemplifying the gating strategy is provided in the Supplementary Information.

# Magnetic resonance imaging

## Experimental design

| | |
|---|---|
| Design type | Indicate task or resting state; event-related or block design. |
| Design specifications | Specify the number of blocks, trials or experimental units per session and/or subject, and specify the length of each trial or block (if trials are blocked) and interval between trials. |
| Behavioral performance measures | State number and/or type of variables recorded (e.g. correct button press, response time) and what statistics were used to establish that the subjects were performing the task as expected (e.g. mean, range, and/or standard deviation across subjects). |

## Acquisition

**Imaging type(s)**  
*Specify: functional, structural, diffusion, perfusion.*

**Field strength**  
*Specify in Tesla*

**Sequence & imaging parameters**  
*Specify the pulse sequence type (gradient echo, spin echo, etc.), imaging type (EPI, spiral, etc.), field of view, matrix size, slice thickness, orientation and TE/TR/flip angle.*

**Area of acquisition**  
*State whether a whole brain scan was used OR define the area of acquisition, describing how the region was determined.*

**Diffusion MRI** ☐ Used ☐ Not used

## Preprocessing

**Preprocessing software**  
*Provide detail on software version and revision number and on specific parameters (model/functions, brain extraction, segmentation, smoothing kernel size, etc.).*

**Normalization**  
*If data were normalized/standardized, describe the approach(es): specify linear or non-linear and define image types used for transformation OR indicate that data were not normalized and explain rationale for lack of normalization.*

**Normalization template**  
*Describe the template used for normalization/transformation, specifying subject space or group standardized space (e.g. original Talairach, MNI305, ICBM152) OR indicate that the data were not normalized.*

**Noise and artifact removal**  
*Describe your procedure(s) for artifact and structured noise removal, specifying motion parameters, tissue signals and physiological signals (heart rate, respiration).*

**Volume censoring**  
*Define your software and/or method and criteria for volume censoring, and state the extent of such censoring.*

## Statistical modeling & inference

**Model type and settings**  
*Specify type (mass univariate, multivariate, RSA, predictive, etc.) and describe essential details of the model at the first and second levels (e.g. fixed, random or mixed effects; drift or auto-correlation).*

**Effect(s) tested**  
*Define precise effect in terms of the task or stimulus conditions instead of psychological concepts and indicate whether ANOVA or factorial designs were used.*

**Specify type of analysis:** ☐ Whole brain ☐ ROI-based ☐ Both

**Statistic type for inference**  
(See Eklund et al. 2016)  
*Specify voxel-wise or cluster-wise and report all relevant parameters for cluster-wise methods.*

**Correction**  
*Describe the type of correction and how it is obtained for multiple comparisons (e.g. FWE, FDR, permutation or Monte Carlo).*

## Models & analysis

| n/a | Involved in the study |
|---|---|
| ☐ | ☐ Functional and/or effective connectivity |
| ☐ | ☐ Graph analysis |
| ☐ | ☐ Multivariate modeling or predictive analysis |

**Functional and/or effective connectivity**  
*Report the measures of dependence used and the model details (e.g. Pearson correlation, partial correlation, mutual information).*

**Graph analysis**  
*Report the dependent variable and connectivity measure, specifying weighted graph or binarized graph, subject- or group-level, and the global and/or node summaries used (e.g. clustering coefficient, efficiency, etc.).*

**Multivariate modeling and predictive analysis**  
*Specify independent variables, features extraction and dimension reduction, model, training and evaluation metrics.*

