## [Peer Review File · Nature Microbiology]

Peer Review Information

Journal: Nature Microbiology

Manuscript Title: Candidatus Alkanophaga archaea from Guaymas Basin hydrothermal vent sediment oxidize petroleum alkanes

Corresponding author name(s): Dr Gunter Wegener, Ms Hanna Zehnle

Decision Letter, initial version:

Message 27th October 2022

:

Dear Ms Zehnle,

Thank you for your patience while your manuscript "Candidatus Alkanophaga archaea from heated hydrothermal vent sediment oxidize petroleum alkanes" was under peer-review at Nature Microbiology. It has now been seen by 3 referees, whose expertise and comments you will find at the end of this email. The reviewers find your work to be of considerable potential interest and impact, and their comments have affirmed the reasons we were editorially interested in your work. As you will see, however, they have raised a number of technical queries, and we agree that addressing these issues would strengthen your conclusions and will be necessary before we can consider publication of the work in Nature Microbiology.

In particular, Reviewer #1 questioned the environmental significance of Ca. Alkanophaga, given that it was only found in the Guaymas Basin and at very low abundance. We are curious if you could extend the reach of the conclusions by searching in other metagenomes? Reviewer #1 also commented on the need for chromatograms to be done with Alkyl-CoM standards, the specificity of the archaeal probes used, and the potential for there to be other hexane degraders at the end of your incubations. Should further experimental data or discussion allow you to address these criticisms, we would be happy to look at a revised manuscript.

We strongly support public availability of data. Please place the data used in your paper into a public data repository, if one exists, or alternatively, present the data as Source Data or Supplementary Information. If data can only be shared on request, please explain why in your Data Availability Statement, and also in the correspondence with your editor. For some data types, deposition in a public repository is mandatory - more information on

our data deposition policies and available repositories can be found at <https://www.nature.com/nature-research/editorial-policies/reporting-standards#availability-of-data>.

Please include a data availability statement as a separate section after Methods but before references, under the heading "Data Availability". This section should inform readers about the availability of the data used to support the conclusions of your study. This information includes accession codes to public repositories (data banks for protein, DNA or RNA sequences, microarray, proteomics data etc...), references to source data published alongside the paper, unique identifiers such as URLs to data repository entries, or data set DOIs, and any other statement about data availability. At a minimum, you should include the following statement: "The data that support the findings of this study are available from the corresponding author upon request", mentioning any restrictions on availability. If DOIs are provided, we also strongly encourage including these in the Reference list (authors, title, publisher (repository name), identifier, year). For more guidance on how to write this section please see: <http://www.nature.com/authors/policies/data/data-availability-statements-data-citations.pdf>

- * If you have not done so already we suggest that you begin to revise your manuscript so that it conforms to our Article format instructions at <http://www.nature.com/nmicrobiol/info/final-submission>. Refer also to any guidelines provided in this letter.

[redacted]

Note: This url links to your confidential homepage and associated information about manuscripts you may have submitted or be reviewing for us. If you wish to forward this e-mail to co-authors, please delete this link to your homepage first.

Nature Microbiology is committed to improving transparency in authorship. As part of our efforts in this direction, we are now requesting that all authors identified as 'corresponding author' on published papers create and link their Open Researcher and Contributor Identifier (ORCID) with their account on the Manuscript Tracking System (MTS), prior to acceptance. This applies to primary research papers only. ORCID helps the scientific community achieve unambiguous attribution of all scholarly contributions. You can create and link your ORCID from the home page of the MTS by clicking on 'Modify my Springer Nature account'. For more information please visit www.springernature.com/orcid.

If you wish to submit a suitably revised manuscript we would hope to receive it within 6 months. If you cannot send it within this time, please let us know. We will be happy to consider your revision, even if a similar study has been accepted for publication at Nature Microbiology or published elsewhere (up to a maximum of 6 months).

Yours sincerely,

[redacted]

Reviewer Expertise:

Referee #1: archaeal alkane metabolism, 'omics, microbial physiology

Referee #2: archaeal alkane metabolism, syntrophy, microbial physiology

Referee #3: archaeal alkane metabolism, archaeal ecology

Reviewer Comments:

Reviewer #1 (Remarks to the Author):

The study of Zehnle et al. described a novel archaeal lineage capable of oxidizing mid-chain alkanes. The archaeal lineage, named *Candidatus Alkanophaga*, activates alkane via formation of alkyl-CoM, similar to short-chain or long-chain alkane activation in other thermophilic archaea (i.e., *Ca. Syntrophoarchaeum* and *Ca. Methanoliparia*). Further oxidation of alkyl-CoM derivatives involves beta-oxidation pathway, cleavage of acetyl-CoA

3and oxidative Wood Ljungdahl pathway. The reducing equivalents yielded during alkane oxidation in *Ca. Alkanophaga* are apparently shuttled to the sulfate-reducing partner via direct interspecies electron transfer. The study is the first rate in reporting archaea thriving on mid-chain alkanes, expanding current knowledge on archaeal physiology and microbial diversity underlying anaerobic alkane oxidation process.

There are several points that need attention.

1. Environmental significance concerning *Ca. Alkanophaga*. The occurrence of *Ca. Alkanophaga* seems to be confined to Guaymas Basin, and even there, the relative abundance of *Ca. Alkanophaga* is less than 0.1%. Such environmental patterns seem to contradict a significant role of *Ca. Alkanophaga* in petroleum degradation.
2. Detection of alkyl-CoM isomers in cell extracts. The identity of the first peak in ion chromatogram of C6-incubated culture extracts, as well as the unique peak in C14-incubated culture remain unclear. Whether those peaks correspond to 2-alkyl-CoM or other isomers requires comparative analyses with alkyl-CoM standards.
3. Amount of alkane derived carbon derived into archaeal biomass. The slightly lower ratio of DIC production to sulfate consumption than theoretical value (1.25-1.30) maybe due to the action of autotrophs in background populations, which consumes CO₂/DIC produced by *Ca. Alkanophaga*. Alternatively, carbon derived from alkane may excrete as dissolved organic carbon, i.e., fatty acids, lowering the DIC production. These may lead to overestimation of alkane-carbon assimilation into archaeal biomass.
4. Substrate specificity of *Ca. Alkanophaga* species. Extend Data Figure 6 indicated that C10-C14 alkanes sustained sulphide production in n-hexane-oxidizing enrichment culture. This seems to be inconsistent with the conclusion that *Ca. A. volatiphilum* cannot degrade alkanes larger than C9. Did the author check the community composition at the end incubation time point of hexane-oxidizing enrichment cultures supplemented with C10, C12 and C14?
5. Pili-based, nanowire-like structures of alkane-oxidizing consortia. High expression level of *pilA* and/or *flaB* in alkane-oxidizing cultures implied direct interspecies electron transfer via pili- or flagella-based nanowire. Did the authors examine the alkane-oxidizing consortia using transmission electron microscope? Are nanowire-like structures detectable in the intercellular space of the consortia?
6. Hybridization should be performed with more specific archaeal probes instead of general Archaeal probe (Arch915). This would provide more accurate information on cell morphology and abundance of *Ca. Alkanophaga*. Currently, it's not clear if the filamentous cells in C6 culture are affiliated to *Ca. Alkanophaga* or other background archaeal populations such as Archaeoglobales, Thermococcales, and Thermoplasmata.

Line 106, if there is clear evidence for misbinning of 16S rRNA gene, should the misplaced contigs be removed from the MAG?

Line 181, it's unclear how propionyl-moiety is completely oxidized into CO₂ via the

4propionylation pathway.

Line 451, it was not clear how binning was performed manually? Are there any specific criteria for manual binning? How do the results differ from binning software?

Reviewer #2 (Remarks to the Author):

Alkanes of 6 to 15 carbons are major components of petroleum deposits in the deep sea and also are abundant of gasoline, diesel and kerosene. Mid-chain length alkanes are toxic. Thus, understanding the fate of these alkanes in an anaerobic environment is thus of great importance. So far, no archaeal cultures have been described that can degrade mid-chain length alkanes. This work describes the two archaea that degrade alkane of 6 to 15 carbons and the mechanisms used for alkane degradation. This work also provides greater understanding of the evolution of hydrocarbon-degrading archaea, suggesting that mid-chain alkane metabolism preceded methane metabolism in the class Syntrophoarchaea. This work also provides critical information needed to determine the distribution of mid-chain-length-alkane degraders which will be important for prediction of global fluxes of these compounds from sediments and their potential effects on the environment. It also increases our understanding of the evolution of hydrocarbon-degrading archaea.

Enrichments degrading mid-chain length alkanes were established from deep sea sediments, which led to the enrichment of two metagenomes (MAG1 and MAG4) with alkane-degrading genes and one metagenome of *Thermodesulfobacteria* (MAG24) that would be the electron-using sulfate-reducing syntrophic partner. Genomic and metabolite analyses showed that alkane activation results in 1-alkyl- or subterminal coenzyme M intermediates showing activation by alkyl-CoM reductases (acr) rather than by fumarate addition, which is often the common mechanism for anaerobic metabolism for alkanes other than methane. MAG 1 and MAG 4 lack the canonical methyl-CoM reductase used by methane and short-chain alkane degraders. The metagenomes contained the genes necessary to completely oxidize the alkane to carbon dioxide and the expression of the requisite genes was confirmed by transcriptional analyses. Each MAG is a new species in a new genus of archaea, *Ca. Alkanophaga*. The syntrophic sulfate-reducing partner is also a new species, *Ca. Thermodesulfobacterium syntrophicum*. Phylogenetic analysis provided new insights into the evolution of Syntrophoarchaea. This work was very challenging and difficult as the alkane degraders had very slow growth rates (13-over 40 days). It is also comprehensive in characterizing the stoichiometry of sulfate reduction coupled to alkane degradation, fluorescent microscopy to show the intimate association of the alkane degrader with its partner and metabolite and transcriptional analyses that confirmed the function of Acr in MAG1 and MAG4 and sulfate reduction in MAG24.

Specific comments:

1. L 229-233 data table 246358. While it is likely that direct electron transfer is involved, H₂ or formate transfer could still be a possibility. The data table indicates that sulfide was still produced even in the presence of molybdate. Molybdate additions should have inhibited sulfate reduction regardless of whether the source of electrons is from H₂,

5formate or conductive pili. In cultures with molybdate, H₂ levels increased above those without molybdate, which would be expected if H₂ is an intermediate in the process. As H₂ levels increase, H₂ production would be inhibited, and some other route may be used. Addition of formate but not H₂, inhibited sulfide production (extended data Fig 4), which may suggest multiple paths of electron flow.

Minor comments:

1. L. 168-171. It is not clear what step the methyltransferase could be used for. The preceding text discusses an aldehyde:ferredoxing reductase, which is a quite different reaction from methyl transfer.
2. Data set 246358 (hydrogen data). The hydrogen concentrations, which are listed in millimolar concentrations, seem very high, 465 mM H₂ after 30 days for C₆ alkane. Should these values be in micromolar rather than in millimolar?
3. Fig 4 and data tables. Is the Fe-ox involved in Etf reoxidation the same as the EMO characterized by Agne et al, 2021. The missing enzymatic link in syntrophic methane formation from fatty acids. Proc Natl Acad Sci U S A 118:e2111682118. If so, it might be best to change the name for consistency.

Michael J. McInerney

Reviewer #3 (Remarks to the Author):

Zehnle and collaborators report the first evidence of the oxidation of alkanes ranging from C₅ to C₁₅, with the implication of Acr complex in the Archaea. In the Alkanophaga, alkane oxidation occurs at a higher temperature compared to what was reported in other archaea. The difference in length of the alkanes used by closely related species is also pretty striking and poses multiple questions. Interestingly, the electron transfer (via DIET) from Alkanophaga species to the sulfate-reducing *Thermodesulfobacterium* may occur through different proteins than those previously reported in syntrophy between alkanotrophic archaea and sulfate reducing bacteria. The fact that multi-carbon preceded one carbon utilization in Syntropharchaeia was already discussed multiple times precedingly. Most the methods and approaches were already successfully employed to study alkane oxidation in Syntropharchaeum, *Ethanoperedens*, *Argoarchaeum* and *Methanoliparum*.

Overall the manuscript reads well, the results are sounds, the methods are clearly described and strong.

I don't have much comments to add.

Line 117: starting by calling them Mcr can be misleading. It's not like it was the first description of an Acr complex.

Lines 155-156: This could be said before explaining the names at lines 148-151

Line 165: why talking about "the (reverse) methanogenesis pathway" here?

6Lines 293-295: This was previously proposed and discussed.

Line 303: This transfer was first described in Borrel et al. 2019 (Wide diversity..).

Line 318: Not clear, what was lost in Alkanophaga? the Borgs, the MHCs, or both. This discussion on Borgs is not very clear.

Author Rebuttal to Initial comments

Reviewer Expertise:

Referee #1: archaeal alkane metabolism, 'omics, microbial physiology

Referee #2: archaeal alkane metabolism, syntrophy, microbial physiology

Referee #3: archaeal alkane metabolism, archaeal ecology

Reviewer Comments:

Reviewer #1 (Remarks to the Author):

The study of Zehnle et al. described a novel archaeal lineage capable of oxidizing mid-chain alkanes. The archaeal lineage, named *Candidatus Alkanophaga*, activates alkane via formation of alkyl-CoM, similar to short-chain or long-chain alkane activation in other thermophilic archaea (i.e., *Ca. Syntrophoarchaeum* and *Ca. Methanoliparia*). Further oxidation of alkyl-CoM derivatives involves beta-oxidation pathway, cleavage of acetyl-CoA and oxidative Wood-Ljungdahl pathway. The reducing equivalents yielded during alkane oxidation in *Ca. Alkanophaga* are apparently shuttled to the sulfate-reducing partner via direct interspecies electron transfer. The study is the first rate in reporting archaea thriving on mid-chain alkanes, expanding current knowledge on archaeal physiology and microbial diversity underlying

anaerobic alkane oxidation process.

Author comment (AC): We thank Reviewer #1 for supporting our study. A point-by-point answer is found below.

There are several points that need attention.

1. Environmental significance concerning *Ca. Alkanophaga*. The occurrence of *Ca. Alkanophaga* seems to be confined to Guaymas Basin, and even there, the relative abundance of *Ca. Alkanophaga* is less than 0.1%. Such environmental patterns seem to contradict a significant role of *Ca. Alkanophaga* in petroleum degradation.

AC: Indeed, the inoculate contained only low numbers of *Ca. Alkanophaga*. Nonetheless, we rapidly enriched this organism from this sediment mix. *Ca. Alkanophaga* is far more abundant in deeper layers of the Guaymas Basin (up to 38% of all reads) as analyzed from publically available metagenomes (see new Extended Data Table 3). These layers are more heated and apparently favor the growth of *Ca. Alkanophaga*. This suggests that *Ca. Alkanophaga* has an important role in heated, hydrocarbon-rich environments.

In addition, many oil-rich and heated environments are very difficult to access and sample, and not many metagenomes from such environments are available. Up until recently, microbial communities in such environments, if at all, were mainly analyzed by 16S rRNA gene amplicon sequencing. We did not find 16S sequences of *Ca. Alkanophaga* in public databases. However, we discovered that the commonly used archaeal primer, Arch915, has a mismatch to both *Ca. Alkanophaga* 16S sequences, which suggests a primer bias against *Ca. Alkanophaga* and could lead to its underrepresentation. This seems to be the case even in the Guaymas Basin, where *Ca. Alkanophaga* sequences do not appear in 16S rRNA libraries. We expect that future metagenomics-based studies in oil-rich environments will disclose the presence of *Ca. Alkanophaga* in other locations.

We modified the text accordingly:

Lines 86-90: “Both MAGs were rare in the original slurry (Extended Data Fig. 1e,f) with relative abundances of $\leq 0.1\%$. This is probably due to the relatively low *in situ* temperatures of the studied sediment (Extended Data Fig. 1d), which included only the upper sediment layer (0-30 cm). Both MAGs are much more abundant in deeper, more heated sediment layers of the

Guaymas Basin³¹, where they recruit up to 39% (MAG 4) and 5% (MAG 1) of raw reads (Extended Data Table 3).”

Lines 335-349: “So far, all *Ca. Alkanophaga* sequences originate from the Guaymas Basin, a thoroughly studied hydrothermal vent area hauling heated fluids rich in alkanes⁷⁶. We suspect two main reasons for this seeming absence in other environments. First, up until recently, microbial community studies have mostly focused on 16S amplification and sequencing, which depends heavily on primer choice⁷⁷. Interestingly, we detected a mismatch of the commonly used archaeal primer Arch915 (5'-GTGCTCCCCGCAATTCCT-3'⁷⁸, mismatch in bold) to the 16S sequences of *Ca. Alkanophaga*. This mismatch could produce an artificial underrepresentation of *Ca. Alkanophaga* sequences in public 16S databases. Second, sequencing data from other environments similar to the Guaymas Basin, i.e. heated oil reservoirs with sulfate supply, remains scarce. Many of these reservoirs are extremely hard to access, being located kilometers deep within the biosphere⁷⁹. In addition to the restricted accessibility, the risk of contamination from the upper biosphere, which might conceal the native community, increases with depth⁷⁹. Still, sampling technologies have greatly improved in recent years, and the focus has shifted from 16S to metagenomics studies, which should facilitate a more accurate molecular characterization of reservoir microorganisms. Thus, future studies may disclose the coexistence and activity of *Ca. Alkanophaga* and *Ca. T. syntrophicum* in other heated, petroleum-rich environments.”

2. Detection of alkyl-CoM isomers in cell extracts. The identity of the first peak in ion chromatogram of C₆-incubated culture extracts, as well as the unique peak in C₁₄-incubated culture remain unclear. Whether those peaks correspond to 2-alkyl-CoM or other isomers requires comparative analyses with alkyl-CoM standards.

AC: To further clarify the identity of the peaks in the cultures, we synthesized three additional standards (2-C₆-CoM, 3-C₆-CoM, and 2-C₁₄-CoM) from the corresponding bromoalkanes (2-bromohexane, 3-bromohexane, and 2-bromotetradecane). We could not produce authentic 3-, or 4-tetradecyl-CoM standards, because the respective precursors (3- and 4-bromotetradecane) were not commercially available.

The new analyses of standards and culture extracts indicated that the main activation products of C₆ were 1-C₆-CoM and 2-C₆-CoM with possibly a small contribution of 3-C₆-CoM (see new Figure 3c). For C₁₄, we observed small peaks in the culture extract corresponding to the retention times of 1-C₁₄-CoM and 2-C₁₄-CoM (Figure 3d). However, the largest peak we

observed in the C_{14} culture appeared at a visibly shorter retention time than both measured standards (Figure 3d). To find out whether this large peak also corresponds to a C_{14} -CoM, we analyzed the C_{14} culture extract additionally in tandem mass spectrometry (MS/MS) mode and produced mass spectra of the fragmentation products of the C_{14} -CoM (m/z 337.1877). From the MS/MS data, we extracted a chromatogram using the exact mass of the C_{14} -thiolate ($C_{14}H_{29}S^-$, m/z 229.1995) (Fig. 3d, bottom panel). The peaks in this chromatogram follow the ones in the EIC produced with the mass of the unfragmented C_{14} -CoM very closely. This suggests that the peaks in the culture extract, including the large one, in the original, unfragmented chromatogram all correspond to C_{14} -CoMs. Since we could not produce >2 - C_{14} -CoM standards, we can only state that C_{14} becomes predominantly activated to >2 - C_{14} -CoM.

3. Amount of alkane derived carbon derived into archaeal biomass. The slightly lower ratio of DIC production to sulfate consumption than theoretical value (1.25-1.30) may be due to the action of autotrophs in background populations, which consumes CO_2 /DIC produced by Ca. Alkanophaga. Alternatively, carbon derived from alkane may excrete as dissolved organic carbon, i.e., fatty acids, lowering the DIC production. These may lead to overestimation of alkane-carbon assimilation into archaeal biomass.

AC: We thank the reviewer for this suggestion. So far, we have not analyzed the organic fraction, but it will be a great addition to the follow-up project.

We corrected the DIC data (shown in Fig. 2c,d, and Supplementary Table 2), where we were missing a factor correcting for the machine efficiency. After recalculating the DIC values, the ratios of DIC production to sulfate reduction became slightly higher, which decreased the estimated assimilation efficiencies to ~10% for the C_6 culture and ~35% for the C_{14} culture. Therefore we modified lines 76-82:

“In two representative cultures, this ratio was slightly lower than that, with 1.21 (± 0.22) in the C_6 culture, and 1.09 (± 0.04) in the C_{14} culture. These values suggest that around 10% (C_6) and 35% (C_{14}) of the carbon released from alkane oxidation is assimilated into biomass (Supplementary Table 2). Since DIC may be removed by autotrophic community members or shuttled into the production of dissolved organic carbon compounds like volatile fatty acids²⁹, actual assimilation efficiencies might be lower than these estimated values. Still, the efficiencies most likely surpass those of methane-oxidizing archaea³⁰, and seem to increase with alkane length.”

4. Substrate specificity of *Ca. Alkanophaga* species. Extended Data Figure 6 indicated that C10-C14 alkanes sustained sulphide production in n-hexane-oxidizing enrichment culture. This seems to be inconsistent with the conclusion that *Ca. A. volatiphilum* cannot degrade alkanes larger than C9. Did the author check the community composition at the end incubation time point of hexane-oxidizing enrichment cultures supplemented with C10, C12 and C14?

AC: We based our conclusions on the long lag times observed when we incubated the C₆ culture with alkanes longer than C₉. In these incubations, sulfide formation appeared only after weeks-months of incubation. Unfortunately, we did not analyze the community compositions again after this experiment.

Therefore we modified the statement: “Substrate tests suggest that *Ca. A. volatiphilum* prefers alkanes <C₁₀, because C₆ culture aliquots immediately produced sulfide when supplied with C₈ and C₉, but exhibited increasing lag times when exposed to alkanes ≥ C₁₀. In contrast, *Ca. A. liquidiphilum* readily degrades all alkanes between C₆ and C₁₅.” (Lines 153-155).

5. Pili-based, nanowire-like structures of alkane-oxidizing consortia. High expression level of pilA and/or flaB in alkane-oxidizing cultures implied direct interspecies electron transfer via pili- or flagella-based nanowire. Did the authors examine the alkane-oxidizing consortia using transmission electron microscope? Are nanowire-like structures detectable in the intercellular space of the consortia?

AC: We performed preliminary TEM analyses of the consortia. The intercellular space seems to contain filament-like structures (see Extended Data Figure 10). To resolve the identity of these structures (i.e. PilA, FlaB, or cytochromes), additional tests such as antibody analyses or protein-targeting TEM are required. Such analyses are beyond the scope of this ending PhD project, but we hope to investigate interactions of both cell types in a follow-up project. We added the text “While further analyses are necessary to confirm that the cells produce conductive nanowires, we observed diffuse structures in the intercellular space that might pertain to such nanowire-like structures (Extended Data Fig. 10)” (lines 259-261) and the method section “Transmission electron microscopy” (lines 633-644).

6. Hybridization should be performed with more specific archaeal probes instead of general Archaeal probe (Arch915). This would provide more accurate information on cell morphology and abundance of *Ca. Alkanophaga*. Currently, it’s not clear if the filamentous cells in C6 culture

are affiliated to *Ca. Alkanophage* or other background archaeal populations such as Archaeoglobales, Thermococcales, and Thermoplasmata.

AC: We thank Reviewer #1 for their request. We developed a specific CARD-FISH probe for *Ca. Alkanophaga* (Aph183) that we successfully applied in a double hybridization with the general bacterial probe EUBI-III. We replaced the previous micrographs produced with general archaeal probes with those made with the *Ca. Alkanophaga*-specific probe Aph183 (Fig. 2c-f).

We modified

- Lines 499-511 of the method section “*In situ* hybridization and microscopy”: “A specific probe was designed to exclusively target the *Ca. Alkanophaga* clade. Therefore, the *Ca. Alkanophaga* 16S rRNA gene sequences were added to the SILVA SSU ref database (version 138.1; <https://www.arb-silva.de/documentation/release-1381/>) using the ARB software¹⁰³. A subtree containing all available ANME-1 16S rRNA gene sequences, plus the two sequences from *Ca. Alkanophaga*, was calculated using RAXML (version 8; <https://cme.h-its.org/exelixis/web/software/raxml/>) with 100 bootstrap replicates, a 50% similarity filter, the GTRGAMMA model, and *Methanocella* as outgroup. The specific probe was designed using the probe design feature with these parameters: length of probe: 19 nucleotides, temperature: 50-100°C, GC content: 50-100%, *E. coli* position: any, max. nongroup hits: 5, min. group hits: 100%. Criteria for candidate probes were: GC content lower than 60%, lowest possible number of matches to non-group species with decreasing temperature, at least one mismatch to non-group species). We ordered a probe that fit these criteria (Aph183) with the sequence 5'-GCATCCAGCACTCCATGG-3' from Biomers (Ulm, Germany).”
- Lines 106-110 of the result section “The enriched archaea activate alkanes with highly transcribed Acrs”: “Visualization of the organisms in the two representative cultures revealed mixed aggregates of archaea of the *Ca. Alkanophagales* clade and bacteria (Fig. 2c-f). In the C₆ culture (Fig. 2c,d), the ratio of *Ca. Alkanophagales* archaea to bacteria appeared higher than in the C₁₄ culture (Fig. 2e,f). The observed associations resemble those of short-chain alkane-oxidizing cultures^{16,17,22}, inferring that the archaea oxidize the alkanes and partner SRB perform sulfate reduction.”

to include the development and application of the specific probe. The images reflect that *Ca. Alkanophaga* are small cells abundant in the aggregates that co-occur with bacterial cells (Fig. 2c-f). By this point, we found mostly aggregates in the C₆ cultures, instead of the filaments detected in the earlier cultures.

12Line 106, if there is clear evidence for misbinning of 16S rRNA gene, should the misplaced contigs be removed from the MAG?

AC: In principle, we agree. Unfortunately, we have already submitted the genome to NCBI and they are scheduled to be published by 9th of February. Therefore, removing the contig containing the second 16S rRNA sequence from this MAG is currently not possible.

Line 181, it's unclear how propionyl-moiety is completely oxidized into CO₂ via the propionylation pathway.

AC: After further consideration, we decided to remove the section regarding the propionylation pathway, because we have too little genomic evidence for this pathway. Therefore, we modified the statement to: "Thus, the fate of the propionyl-CoA remains, for the moment, unclear." (Lines 181-182)

Line 451, it was not clear how binning was performed manually? Are there any specific criteria for manual binning? How do the results differ from binning software?

AC: We agree, and detailed the manual binning procedure in the method section "Short-read DNA data analysis" (lines 438-444): "The contigs database was inspected in the *anvi'o* interactive interface, which clusters the contigs hierarchically based on sequence composition and differential coverage, thereby indicating their relatedness to each other⁸⁸. Binning was performed manually in the interface by clicking branches of the dendrogram in the center of the interface, and using the GC content, mean coverage in the samples, gene taxonomy, and real-time statistics on completion and redundancy based on single-copy core genes as guides. The dendrogram branches were followed systematically in a counterclockwise direction in order to achieve the maxima number of bins."

The results obtained with manual ("human-guided") binning are almost identical to results obtained with the binning software MetaBAT (v.2.12.1; <https://bitbucket.org/berkeleylab/metabat/src/master/>) shown here for comparison:

MetaBAT was run with default settings, using the contig depth in the samples' BAM files and the assembly fasta as input files. MAGs were refined in the anvio interactive interface, aiming to decrease redundancy. Final MAG quality was assessed with CheckM. MAGs with completeness > 50% and redundancy < 10% were kept. Relative abundance of the MAGs was calculated with CoverM.

Reviewer #2 (Remarks to the Author):

Alkanes of 6 to 15 carbons are major components of petroleum deposits in the deep sea and also are abundant of gasoline, diesel and kerosene. Mid-chain length alkanes are toxic. Thus, understanding the fate of these alkanes in anaerobic environment is thus of great importance. So far, no archaeal cultures have been described that can degrade mid-chain length alkanes. This work describes the two archaea that degrade alkane of 6 to 15 carbons and the mechanisms used for alkane degradation. This work also provides greater understanding of the evolution of hydrocarbon-degrading archaea, suggesting that mid-chain alkane metabolism proceeded methane metabolism in the class Syntrophoarchaeia. This work also provide critical information needed to determine the distribution of mid-chain-length-alkane degraders which will is important for prediction global fluxes of these compounds from sediments and their potential effects on the environment. It also increases our understanding of the

14evolution of hydrocarbon-degrading archaea.

Enrichments degrading mid-chain length alkanes were established from deep sea sediments, which led to the enrichment of two metagenomes (MAG1 and MAG4) with alkane-degrading genes and one metagenome of Thermodesulfobacteria (MAG24) that would be the electron-using sulfate-reducing syntrophic partner. Genomic and metabolite analyses showed that alkane activation results in 1-alkyl- or subterminal coenzyme M intermediates showing activation by alkyl-CoM reductases (acr) rather than by fumarate addition, which is often the common mechanism for anaerobic metabolism for alkanes other than methane. MAG 1 and MAG 4 lack the canonical methyl-CoM reductase used by methane and short-chain alkane degraders. The metagenomes contained the genes necessary to completely oxidize the alkane to carbon dioxide and the expression of the requisite genes was confirmed by transcriptional analyses. Each MAG is a new species in a new genus of archaea, *Ca. Alkanophaga*. The syntrophic sulfate-reducing partner is also new species, *Ca. Thermodesulfobacterium syntrophicum*. Phylogenetic analysis provided new insights into the evolution of Syntrophoarchaea. This work was very challenging and difficult as the alkane degraders had very slow growth rates (13-over 40 days). It is also comprehensive in characterizing the stoichiometry of sulfate reduction coupled to alkane degradation, fluorescent microscopy to show the intimate association of the alkane degrader with its partner and metabolite and transcriptional analyses that confirmed the function of Acr in MAG1 and MAG4 and sulfate reduction in MAG24.

AC: We thank Reviewer #2 for acknowledging our cultivation and analytical challenges, the valuable comments and their support of this study.

Specific comments:

1. L 229-233 data table 246358. While it is likely that direct electron transfer is involved, H₂ or formate transfer could still be a possibility. The data table indicates that sulfide was still produced even in the presence of molybdate. Molybdate additions should have inhibited sulfate reduction regardless of whether the source of electrons is from H₂, formate or conductive pili. In cultures with molybdate, H₂ levels increased above those without molybdate, which would be expected if H₂ is an intermediate in the process. As H₂ levels increase, H₂ production would be inhibited, and some other route may be used. Addition of

15formate but not H₂, inhibited sulfide production (extended data Fig 4), which may suggest multiple paths of electron flow.

AC: Reviewer #2 is correct. Indeed, molybdate blocked sulfide formation also in our experiment. In Supplementary Table 6 (referred to as data table 246358), the “sulfide” column displays a theoretical sulfide production rate, which we extrapolated from the sulfide production measured in the 21 days prior to the addition of molybdate. We changed the column title to “Sulfide produced - theoretical” for clarification. From the theoretical sulfide production, we calculated the theoretical production of H₂ during blockage of sulfate reduction under the assumption that H₂ is the sole electron carrier during the oxidation of hexane ($C_6H_{14} + 6 H_2O + 12 H^+ \rightarrow 6 CO_2 + 19 H_2$ | $SO_4^{2-} + 4 H_2 + H^+ \rightarrow HS^- + 4 H_2O$) or tetradecane ($C_{14}H_{30} + 14 H_2O + 28 H^+ \rightarrow 14 CO_2 + 43 H_2$ | $SO_4^{2-} + 4 H_2 + H^+ \rightarrow HS^- + 4 H_2O$). Then, we calculated the percentage of actually measured H₂ production compared to this theoretical production. Thereby, we found only traces of the theoretical H₂ production (max. 2.4%). We agree with the reviewer that other soluble electron carriers remain possible intermediates (lines 265-267) “It remains a possibility that a small fraction of electrons are transferred via soluble intermediates like hydrogen. Such a combination of DIET with diffusion-based electron transport was recently shown to be energetically favorable for syntrophic consortia⁶¹.”

Minor comments:

1. L. 168-171. It is not clear what step the methyltransferase could be used for. The preceding text discusses an aldehyde:ferredoxing reductase, which is a quite different reaction from methyl transfer.

AC: We agree, we improved the description of the potential role of this enzyme (lines 170-172) for clarification: “For *Ca. Syntrophoarchaeum*, it was proposed that a highly transcribed methyltransferase transfers the alkyl moieties from CoM to CoA¹⁸; however, it is highly unlikely that this enzyme is capable of utilizing the large alkane substrates consumed by *Ca. Alkanophaga*.”

2. Data set 246358 (hydrogen data). The hydrogen concentrations, which are listed in millimolar concentrations, seem very high, 465 mM H₂ after 30 days for C₆ alkane. Should these values be in micromolar rather than in millimolar?

AC: We thank the reviewer for this remark. Indeed, our calculation had a major bug. We fixed it, and the H₂ production rates concentrations are now in micromole. The values of the percentage of sulfide production that is explained by H₂ production changed slightly as a result, however they remain at maxima of ~2% for the C₆ culture and ~0.9% for the C₁₄ culture (the values in line 233 were changed to include this slight change).

3. Fig 4 and data tables. Is the Fe-ox involved in Etf reoxidation the same as the EMO characterized by Agne et al, 2021. The missing enzymatic link in syntrophic methane formation from fatty acids. Proc Natl Acad Sci U S A 118:e2111682118. If so, it might be best to change the name for consistency.

AC: Both MAGs 1 and 4 encode several Fe-S oxidoreductases, however, all have very low identities to the EMO protein (SYN_02638) reported in Agne *et al* (percent identity ranging from 22.2-48.1% according to BLASTp).

Reviewer #3 (Remarks to the Author):

Zehnle and collaborators report the first evidence of the oxidation of alkanes ranging from C₅ to C₁₅, with the implication of Acr complex in the Archaea. In the Alkanophaga, alkane oxidation occurs at a higher temperature compared to what was reported in other archaea. The difference in length of the alkanes used by closely related species is also pretty striking and poses multiple questions. Interestingly, the electron transfer (via DIET) from Alkanophaga species to the sulfate-reducing Thermodesulfobacterium may occur through different proteins than those previously reported in syntrophy between alkanotrophic archaea and sulfate reducing bacteria. The fact that multi-carbon preceded one carbon utilization in Syntropharchaeia was already discussed multiple times precedingly. Most the methods and approaches were already successfully employed to study alkane oxidation in Syntropharchaeum, Ethanoperedens, Argoarchaeum and Methanoliparum.

Overall the manuscript reads well, the results are sounds, the methods are clearly described and strong.

I don't have much comments to add.

AC: We thank Reviewer #3 for his comments and the support of the manuscript. Based on the Reviewers remark we shortened the introduction, to be less redundant in an important aspect.

Line 117: starting by calling them Mcr can be misleading. It's not like it was the first description of an Acr complex.

AC: We agree, the term Acr is now quite established. We modified the lines 113-114 to: "Instead, both MAGs encode three Acrs consisting of the three subunits AcrA, AcrB, and AcrG".

Lines 155-156: This could be said before explaining the names at lines 148-151.

AC: The order of these statements was switched as suggested, and the whole section (lines 147-152) was changed upon the discovery that based on AAI, *Ca.* Alkanophaga do not constitute a separate family as stated in the previous version of the manuscript.

Line 165: why talking about "the (reverse) methanogenesis pathway" here?

AC: We changed "reverse methanogenesis pathway" to "Wood-Ljungdahl pathway" (now in line 166).

Lines 293-295: This was previously proposed and discussed.

AC: We changed the text in lines 296-297 to: "Our study supports the previously established hypothesis that multi-carbon alkane metabolism likely preceded methanotrophy in the Syntrophoarchaea^{20,67}" and added Wang *et al* 2021 (DOI: 10.1126/sciadv.abj1453) as a reference.

[Line 303: This transfer was first described in Borrel *et al.* 2019 (Wide diversity..).

We added the work of Borrel *et al* 2019 as a reference (now in line 306-307).

Line 318: Not clear, what was lost in Alkanophaga? the Borgs, the MHCs, or both. This discussion on Borgs is not very clear.

AC: The idea is that the Borgs, which encoded the multiple MHCs, were lost in *Ca. Alkanophaga*. We changed the lines 322-323 for clarification: "In a recent study, giant extrachromosomal elements named Borgs, many of which carried clusters of MHCs, were reconstructed from methane-oxidizing *Methanoperedens* (ANME-2d) archaea⁷⁴. It is therefore possible that MHCs in the Syntrophoarchaea are encoded on such a Borg, and that such a Borg which included the MHCs was lost by *Ca. Alkanophaga*. This could explain why all MHCs are absent in *Ca. Alkanophaga*."

Decision Letter, first revision:

Message: Our ref: NMICROBIOL-22092359A

3rd April 2023

Dear Dr. Zehnle,

Thank you for submitting your revised manuscript "Candidatus *Alkanophaga* archaea from heated hydrothermal vent sediment oxidize petroleum alkanes" (NMICROBIOL-22092359A). It has now been seen by the original referees and their comments are below. The reviewers find that the paper has improved in revision, and therefore we'll be happy in principle to publish it in Nature Microbiology, pending minor revisions to satisfy the referees' final requests and to comply with our editorial and formatting guidelines.

Thank you again for your interest in Nature Microbiology Please do not hesitate to contact me if you have any questions.

Sincerely,
[redacted]

Reviewer #1 (Remarks to the Author):

The authors have adequately addressed all my comments! I would support the publication

19of this great paper in Nature Microbiology.

Song-Can Chen

Reviewer #2 (Remarks to the Author):

The authors added additional information on the distribution of the metagenomes which show higher abundances in lower hotter sediments, added mass spectrometry and microscopic analyses and clarified some statements which I believe answered all of the questions and comments of the reviewers.

Reviewer #3 (Remarks to the Author):

Zehnle and collaborators answered my comments and added important additional results. I have no additional comments.

Final Decision Letter:

Me 28th April 2023

sa

e: Dear Ms Zehnle,

I am pleased to accept your Article "Candidatus Alkanophaga archaea from Guaymas Basin hydrothermal vent sediment oxidize petroleum alkanes" for publication in Nature Microbiology. Thank you for having chosen to submit your work to us and many congratulations.

Due to the importance of these deadlines, we ask you please us know now whether you will be difficult to contact over the next month. If this is the case, we ask you provide us with the

20contact information (email, phone and fax) of someone who will be able to check the proofs on your behalf, and who will be available to address any last-minute problems.

Acceptance of your manuscript is conditional on all authors' agreement with our publication policies (see <https://www.nature.com/nmicrobiol/editorial-policies>). In particular your manuscript must not be published elsewhere and there must be no announcement of the work to any media outlet until the publication date (the day on which it is uploaded onto our website).

Please note that *Nature Microbiology* is a Transformative Journal (TJ). Authors may publish their research with us through the traditional subscription access route or make their paper immediately open access through payment of an article-processing charge (APC). Authors will not be required to make a final decision about access to their article until it has been accepted. [Find out more about Transformative Journals](https://www.springernature.com/gp/open-research/transformative-journals)

Authors may need to take specific actions to achieve [compliance](https://www.springernature.com/gp/open-research/funding/policy-compliance-faqs) with funder and institutional open access mandates. If your research is supported by a funder that requires immediate open access (e.g. according to [Plan S principles](https://www.springernature.com/gp/open-research/plan-s-compliance)) then you should select the gold OA route, and we will direct you to the compliant route where possible. For authors selecting the subscription publication route, the journal's standard licensing terms will need to be accepted, including [self-archiving policies](https://www.nature.com/nature-portfolio/editorial-policies/self-archiving-and-license-to-publish). Those licensing terms will supersede any other terms that the author or any third party may assert apply to any version of the manuscript.

You can now use a single sign-on for all your accounts, view the status of all your manuscript

submissions and reviews, access usage statistics for your published articles and download a record of your refereeing activity for the Nature journals.

With kind regards,

[redacted]

P.S. Click on the following link if you would like to recommend Nature Microbiology to your librarian <http://www.nature.com/subscriptions/recommend.html#forms>

** Visit the Springer Nature Editorial and Publishing website at http://editorial-jobs.springernature.com?utm_source=ejp_NMicro_email&utm_medium=ejp_NMicro_email&utm_campaign=ejp_NMicro for more information about our career opportunities. If you have any questions please click [here](mailto:editorial.publishing.jobs@springernature.com).